# Gradient-free training of recurrent neural networks for low dimensional data

## Abstract

Recurrent neural networks are a successful neural architecture for many time-dependent problems, including time series analysis, forecasting, and modeling of dynamical systems. Training such networks with backpropagation through time is a notoriously difficult problem because their loss gradients tend to explode or vanish. In this contribution, we introduce a computational approach to construct all weights and biases of a recurrent neural network without using gradient-based methods. The approach is based on a combination of random feature networks and Koopman operator theory for dynamical systems. The hidden parameters of a single recurrent block are sampled at random, while the outer weights are constructed using extended dynamic mode decomposition. This approach alleviates all problems with backpropagation commonly related to recurrent networks. The connection to Koopman operator theory also allows us to start using results in this area to analyze recurrent neural networks. In computational experiments on time series, forecasting for chaotic dynamical systems, and control problems, as well as on weather data, we observe that the training time and forecasting accuracy of the recurrent neural networks we construct are improved when compared to commonly used gradient-based methods.

## 1 Introduction

Recurrent neural networks (RNNs) are notoriously difficult to train because their loss gradients backpropagated in time tend to saturate or diverge during training, commonly referred to as the Exploding and Vanishing Gradient Problem (EVGP) (Pascanu et al., 2013; Schmidt et al., 2019). To alleviate these problems and improve the computational load of training such networks, we consider a model that completely avoids iterative gradient-descent optimization. We propose to sample the hidden layer parameters of the RNN at random, before solving for the outer linear layer by least-squares methods. Our first major contribution is that we consider data-dependent distributions for the weights and biases, which improves upon the accuracy and interpretability compared to data-agnostic distributions. Our second major contribution adds more structure to the outer layer and improve the models performance even further. We achieve this by connecting RNNs and the Koopman operator (cf. Korda & Mezić (2018a)) through a linear state-space model in a higher-dimensional space, i.e.,

$$
\underbrace{\begin{aligned} \boldsymbol{h}_t &= F(\boldsymbol{h}_{t-1}, \boldsymbol{x}_t) \\ \boldsymbol{y}_t &= g(\boldsymbol{h}_t) \end{aligned}}_{\text{original non-linear system}} \leftrightarrow \underbrace{\begin{aligned} \boldsymbol{z}_t &= K\boldsymbol{z}_{t-1} + B\boldsymbol{x}_t \\ \boldsymbol{y}_t &= V\boldsymbol{z}_t \end{aligned}}_{\text{linear (Koopman) state-space model}} \leftrightarrow \underbrace{\begin{aligned} \boldsymbol{z}_t &= K\mathcal{F}(\boldsymbol{h}_{t-1}) + B\mathcal{G}(\boldsymbol{x}_t) \\ \boldsymbol{h}_t &= C\boldsymbol{z}_t \\ \boldsymbol{y}_t &= V\boldsymbol{z}_t \end{aligned}}_{\text{non-linear RNN}} .
$$

(1)

Here the matrix $K$ maps states from $\boldsymbol{z}_{t-1}$ to $\boldsymbol{z}_t$, with the input $\boldsymbol{x}_t$ affecting this dynamic through the matrix $B$, and the observations are related to the states through the linear map $V$.

By using a non-linear neural network to map the low-dimensional state $\boldsymbol{h}$ into a higher dimension, through $\mathcal{F}(\boldsymbol{h}) := \sigma(W\boldsymbol{h} + \boldsymbol{b})$, and, similarly, to map the input $\boldsymbol{x}$ to a high-dimensional input $\mathcal{G}(\boldsymbol{x}) = \sigma_x(W_x\boldsymbol{x} + \boldsymbol{b}_x)$, turns the linear SSM into a non-linear recurrent neural network. The final model is illustrated in Figure 1. This connection gives structure and interpretability to the outer layer of RNNs, as well as tools to analyze the RNNs stability through the spectral properties of the Koopman operator.

Figure 1: Illustration of the components of one recurrent block we construct in the paper. The state $z_{t-1}$ enters on the left, and is processed through matrix $C$ and the neural network $\mathcal{F} = \sigma(W \cdot + \boldsymbol{b})$. We then advance in time to $z_t$, using the Koopman matrix $K$ and the processed control inputs.

The combination of sampling hidden layers and applying Koopman theory **alleviates all issues related to the backpropagation of gradients** and **connects the idea of recurrent architectures to the Koopman operator** of the underlying dynamical system. The latter allows us to **prove convergence results** about the approximation quality w.r.t. the number of neurons in $\mathcal{F}$. We **demonstrate the performance** of this approach on several challenging examples with synthetic and real data, and show comparable results to networks trained with backpropagation. Finally, we touch upon the models current challenges such as applying it to high dimensional data.

## 2 RELATED WORK

*Exploding and vanishing gradients.* For all major types of RNNs, including LSTMs and GRUs, the dynamics and loss gradients of RNNs are closely linked. If the RNN dynamics converge to a stable fixed point or cycle, loss gradients will remain bounded, but they may vanish (Mikhaeil et al., 2022). Yet, established remedies (Hochreiter & Schmidhuber, 1997; Schmidt et al., 2019) can be used to effectively prevent their gradients from vanishing. However, in chaotic dynamics, gradients invariably explode, posing a challenge that cannot be mitigated through RNN architectural adjustments, regularization, or constraints; instead, it necessitates addressing the problem during the training process (Mikhaeil et al., 2022). Bifurcations may also contribute to sudden jumps in loss observed during RNN training, potentially hindering the training process severely (Doya et al., 1992; Eisenmann et al., 2023). In Eisenmann et al. (2023), it has been demonstrated that specific bifurcations in ReLU-based RNNs are always associated with EVGP during training. Therefore, to harness the full potential of RNNs, the training algorithm needs careful design to tackle challenges posed by bifurcations and the possible emergence of EVGP.

*Curse of memory.* The existence of long-term memory adversely affects the learning process of RNNs (Bengio et al., 1994; Hochreiter et al., 2001; Li et al., 2021). This negative impact is captured by the concept of the "curse of memory", which states that when long-term memory is present in the data, approximating relationships demands an exponentially large number of neurons, resulting in a significant slowdown in learning dynamics. Specifically, when the target relationship includes long-term memory, both the approximation and optimization of the learning process become very challenging (Li et al., 2021).

*Loss function for chaotic and multistable dynamics reconstruction.* To effectively train RNNs and evaluate reconstruction, it is crucial to carefully choose a proper loss function. The Mean Squared Error (MSE) is a commonly used loss function for reconstruction tasks. MSE is derived under the assumption of Gaussian noise, and it may not be the most appropriate choice when dealing with chaotic systems or multistable dynamics, where the underlying noise characteristics may deviate from Gaussian distribution assumptions. It is not suitable as a test loss for chaotic dynamical systems due to their unpredictable behavior and abrupt changes (Wood, 2010). In multistable systems, where there are multiple stable states, MSE may struggle to distinguish between these states. The loss function may not adequately penalize deviations between different attractors, leading to a less accurate reconstruction of the system's multistable behavior. Despite proposed alternatives, challenges persist, and an optimal loss function for reconstructing chaotic or multistable dynamics is still lacking (Ciampiconi et al., 2023).

*Interpretability deficiency.* Deep learning models are commonly regarded as "black boxes", and existing methods to comprehend the decision-making processes of RNNs offer restricted explanations or rely on local theories. The lack of theory behind analyzing the training algorithms of RNNs, as

well as the training process itself, are outstanding open questions in the field (Redman et al., 2023) beyond models for linear systems (Datar et al., 2024).

*Randomly choosing the internal network parameters.* The general idea of randomly choosing the internal parameters of neural networks is studied in random feature models (including deep architectures).Barron (1993); Rahimi & Recht (2008) developed the basic theory, and Gallicchio & Scardapane (2020) provide a review. Reservoir computing (also called echo-state networks, cf. Jaeger & Haas (2004)) is potentially the closest idea to what we are proposing here. In a reservoir computer, the internal weights are also randomly sampled, and a recurrent, time-delayed model is constructed to approximate a given dynamical system. This type of architecture has been used successfully to model chaotic systems (cf. Pathak et al. (2018); Gauthier et al. (2021)). While there are similarities to the recurrent architecture (cf. Lukoševičius & Jaeger (2009)), the concept, as well as the architecture of a reservoir computer, is often treated separately from classic recurrent neural networks that are trained with backpropagation-in-time. In our work, we directly compute all parameters of classical recurrent neural networks without the time-delay component present in reservoir computers.

*Koopman operator theory.* The Koopman operator is an object associated to every dynamical system. It evolves observables of the state of the system in time. This evolution is linear, which is the main reason the operator is employed and studied extensively for modeling dynamical systems (Mezić, 2005; 2013; Korda & Mezić, 2018b). Many numerical approximation algorithms exist (Schmid, 2010; Williams et al., 2015a; Li et al., 2017; Mezic, 2020; Schmid, 2022). The dictionary for the approximation of the Koopman operator has been constructed with neural networks using gradient descent (Li et al., 2017) and using random features (Salam et al., 2022). Reservoir computing has also been related to Koopman operator approximation by Bollt (2021); Gulina & Mauroy (2020). To our knowledge, the relation of the Koopman operator to the weight matrices of recurrent neural networks has not been observed before. This is what we discuss in this work. We also provide a data-dependent probability distribution for the hidden parameters, which is the strongest deviation from the data-agnostic distributions (e.g., normal, uniform) typically used in reservoir computing.

## 3 Mathematical framework

We start by defining a general framework of recurrent neural networks (RNNs) and the underlying dynamical system before introducing sampling and its connection to the Koopman operator. Let $\mathcal{X} \subseteq \mathbb{R}^{d_x}$ be an input space, $\mathcal{Y} \subseteq \mathbb{R}^{d_y}$ an output space, and $\mathcal{H} \subseteq \mathbb{R}^{d_h}$ a state space. We assume that these spaces are associated with the measures $\mu_x$, $\mu_y$, and $\mu_h$, respectively. The underlying dynamical system is then defined through the evolution operator $F$, where we may be working in an uncontrolled system $\boldsymbol{h}_t = F(\boldsymbol{h}_{t-1})$ or a controlled system $\boldsymbol{h}_t = F(\boldsymbol{h}_{t-1}, \boldsymbol{x}_t)$. We also denote the input dataset $X = [\boldsymbol{x}_1, \boldsymbol{x}_2, \ldots, \boldsymbol{x}_N]$ and the dataset of observations $Y = [\boldsymbol{y}_1, \boldsymbol{y}_2, \ldots, \boldsymbol{y}_N]$. In this paper we are interested in recurrent neural networks modelling dynamical systems where the state is observable, and we therefore usually assume access to the dataset $H = [\boldsymbol{h}_1, \ldots, \boldsymbol{h}_N]$ as well as its copy after one time step $H' = [\boldsymbol{h}'_1, \ldots, \boldsymbol{h}'_N]$, where $\boldsymbol{h}'_n = F(\boldsymbol{h}_n)$, $n \in \{1, \ldots, N\}$.

We denote activation functions as $\sigma \colon \mathbb{R} \to \mathbb{R}$, where we are mainly working with $\sigma = \tanh$ in this paper, as it is an analytic function and connects to SWIM (Bolager et al., 2023). Other functions such as ReLU are also a valid choice. The following definition outlines the models we consider.

**Definition 1.** *Let $W_h \in \mathbb{R}^{M \times d_h}$, $W_x \in \mathbb{R}^{\hat{M} \times d_x}$, $\boldsymbol{b}_h \in \mathbb{R}^M$, $\boldsymbol{b}_x \in \mathbb{R}^{\hat{M}}$, $C_h \in \mathbb{R}^{d_h \times M}$, $C_x \in \mathbb{R}^{d_h \times \hat{M}}$, and $V \in \mathbb{R}^{d_y \times d_h}$. For time step $t$ and $\boldsymbol{h}_0 \in \mathcal{H}$, we define a recurrent neural network (RNN) by*

$$\boldsymbol{h}_t = \sigma_{hx}(C_h \, \sigma(W_h \, \boldsymbol{h}_{t-1} + \boldsymbol{b}_h) + C_x \, \sigma(W_x \, \boldsymbol{x}_t + \boldsymbol{b}_x) + \boldsymbol{b}_{hx}), \tag{2}$$
$$\boldsymbol{y}_t = V \boldsymbol{h}_t. \tag{3}$$

*Remark* 1. For completeness we have added $\sigma_{hx}$ as an arbitrary activation function. We choose to set $\sigma_{hx}$ as the identity function to let us solve for the last linear layer in the procedure described below. Other activation functions such as the logit is possible as well.

The classical way to train this type of RNN is through iterative backpropagation, which suffers the aforementioned issues such as EVGP and high computational complexity. We instead start by sampling the hidden layer parameters to circumvent backpropagation, as explained next.

### 3.1 SAMPLING RNN

Sampled neural networks are neural networks where the parameters of the hidden layers are sampled from some distribution, and the last linear layer is either sampled or, more typically, solved through a linear solver. Following Bolager et al. (2023), we sample the weights and biases of the hidden layers of both $F_{\mathcal{H}}$ and $F_{\mathcal{X}}$ by sampling pairs of points from the domain $\mathcal{H}$ and $\mathcal{X}$, and construct the weights and biases from these pairs of points. One then solves a linear (general) regression problem at the end for the last linear layer that maps to the next state. Concretely, let $\mathbb{P}_{\mathcal{H}}$ and $\mathbb{P}_{\mathcal{X}}$ be probability distributions over $\mathcal{H}^2$ and $\mathcal{X}^2$ respectively. For each neuron in the hidden layer of $F_{\mathcal{H}}$, sample $(\boldsymbol{h}^{(1)}, \boldsymbol{h}^{(2)}) \sim \mathbb{P}_{\mathcal{H}}$ and set the weight $w$ and the bias $b$ of said neuron to

$$\boldsymbol{w} = s_1 \frac{\boldsymbol{h}^{(2)} - \boldsymbol{h}^{(1)}}{\|\boldsymbol{h}^{(2)} - \boldsymbol{h}^{(1)}\|^2}, \quad b = -\langle \boldsymbol{w}, \boldsymbol{h}^{(1)} \rangle + s_2, \tag{4}$$

where $\|\cdot\|$ and $\langle \cdot, \cdot \rangle$ are typically the Euclidean norm and inner product, and $s_1, s_2 \in \mathbb{R}$ are constants. Repeating the same procedure for all neurons in both $F_{\mathcal{H}}$ and $F_{\mathcal{X}}$. As we stick to networks with one hidden layer in this paper, we ignore the multilayer sampling here and direct the reader to Bolager et al. (2023) for the full sample and construction procedure for an arbitrary number of hidden layers.

This sampling technique adapts the weights and biases to the underlying domain and constructs weights with direction along the data (see Appendix C for an example how this can be used to interpret the resulting network). Empirically, this has shown to be an improvement over using data-agnostic distributions such as the standard Gaussian one. One can choose arbitrary probability distributions as $\mathbb{P}_{\mathcal{H}}$ and $\mathbb{P}_{\mathcal{X}}$, with uniform distribution being a common choice. For the supervised setting, Bolager et al. (2023) also proposed a sampling distribution whose density captures the steepest gradients of the target function. For this paper, we sample with densities $p_{\mathcal{H}}$ and $p_{\mathcal{X}}$ proportional to

$$p_{\mathcal{H}} \propto \frac{\|F(\boldsymbol{h}^{(2)}) - F(\boldsymbol{h}^{(1)})\|}{\|\boldsymbol{h}^{(2)} - \boldsymbol{h}^{(1)}\|}, \quad p_{\mathcal{X}} \propto 1,$$

respectively. Once the weights and biases are sampled, we must solve a general regression problem

$$[C_h, C_x] = \underset{\hat{C}_h, \hat{C}_x}{\arg\min} \sum_{n=1}^{N} \|(\hat{C}_h \, \sigma(W_h \, \boldsymbol{h}_n + \boldsymbol{b}_h) + \hat{C}_x \, \sigma(W_x \, \boldsymbol{x}_n + \boldsymbol{b}_x) + \boldsymbol{b}_{hx}) - \boldsymbol{h}'_n\|^2. \tag{5}$$

To summarize, we define a *sampled RNN* as a model that is constructed by sampling weights of the hidden layer of the RNN and subsequently solving the regression problem in Equation (5).

### 3.2 INVOLVING THE KOOPMAN OPERATOR

We already introduced the network $\mathcal{F}_M \colon \mathbb{R}^{d_h} \to \mathbb{R}^M = \sigma(W_h \cdot + \boldsymbol{b}_h)$, where $M$ is the number of neurons in its single hidden layer, and likewise with $\mathcal{G}_{\hat{M}} \colon \mathbb{R}^{d_x} \to \mathbb{R}^{\hat{M}} = \sigma(W_x \cdot + \boldsymbol{b}_x)$. We then project down to $\mathbb{R}^{d_h}$ by applying $C_h$ and $C_x$ respectively. To add more structure and interpretability to the matrices $C_h$ and $C_x$, we will set $C_h = CK$ and $C_x = CB$, where $K \in \mathbb{R}^{M \times M}$, $B \in \mathbb{R}^{M \times \hat{M}}$, and $C \in \mathbb{R}^{d_h \times M}$. We then end up with the function

$$\boldsymbol{h}_t = C\boldsymbol{z}_t = C(K\sigma(W_h \, \boldsymbol{h}_{t-1} + \boldsymbol{b}_h) + B\sigma(W_x \, \boldsymbol{x}_t + \boldsymbol{b}_x)). \tag{6}$$

Note that if the output $\boldsymbol{y}$ differs from $\boldsymbol{h}$, we take advantage of the high dimensionality and rather set $\boldsymbol{y}_t = V\boldsymbol{z}_t$ than first projecting down by $C$. This completes the setting from Equation (1).

The reason for the splitting of matrices $C_h$ and $C_x$ is the following: the hidden layers $\mathcal{F}_M$ and $\mathcal{G}_{\hat{M}}$ map their respective input to a higher dimensional space. In a higher dimensional space, the possibly nonlinear evolution described by $F$ becomes more and more linear (Korda & Mezić, 2018b). This evolution is then captured by $K$ and $B$ before we map down to the state space through $C$. This also allows us to connect $K$ and $B$ to the Koopman theory applied to the dynamical system. Given a suitable functional space $\mathcal{F}$, the Koopman operator $\mathcal{K} \colon \mathcal{F} \to \mathbb{R}$ is defined as

$$[\mathcal{K}\phi](\boldsymbol{h}) = (\phi \circ F)(\boldsymbol{h}), \quad \phi \in \mathcal{F}.$$

The Koopman operator captures the evolution of the dynamical systems in the function space $\mathcal{F}$ and not in the state space itself. In most cases, this makes the operator infinite-dimensional, but in return,

it is a linear operator. For an introduction to the Koopman operator and surrounding theory, see Appendix A. The matrices $K$ and $B$ can then be seen as an approximation of the Koopman operator, and the intuition is that choosing $M$ large enough means we can capture the linear evolution before we map down to the state space.

The matrices $W_h$ and $W_x$ are found by sampling each row i.i.d. from $\mathbb{P}_h$ and $\mathbb{P}_x$ respectively. To estimate $C$, $K$, and $B$, we use extended dynamic mode decomposition (EDMD) — a classical method to find a finite-dimensional approximation of the Koopman operator. We give a very brief description of EDMD here and give a more thorough introduction in Appendix A.1. In the uncontrolled setting, this method picks out a dictionary $\mathcal{F}_M = \{\psi_1, \ldots, \psi_M \colon \psi_i \colon \mathcal{H} \to \mathbb{R} \in \mathcal{F}\}$ and estimates the Koopman operator $\mathcal{K}$ using the data $H, H'$ by minimizing

$$K = \arg\min_{\tilde{K} \in \mathbb{R}^{M \times M}} \sum_{n=1}^{N} \|\mathcal{F}_M(\boldsymbol{h}'_n) - \tilde{K}\mathcal{F}_M(\boldsymbol{h}_n)\|, \quad \boldsymbol{h}'_n \in H', \boldsymbol{h}_n \in H,$$

where $\mathcal{F}_M(\boldsymbol{h}) = [\phi_1(\boldsymbol{h}), \ldots, \phi_M(\boldsymbol{h})]^{\mathsf{T}}$. Letting $\mathcal{F}_M(H) = [\mathcal{F}_M(\boldsymbol{h}_1), \ldots, \mathcal{F}_M(\boldsymbol{h}_N)] \in \mathbb{R}^{M \times N}$ and $\mathcal{F}_M(H') = [\mathcal{F}_M(\boldsymbol{h}'_1), \ldots, \mathcal{F}_M(\boldsymbol{h}'_N)] \in \mathbb{R}^{M \times N}$, the approximation can then be written as

$$K = \mathcal{F}_M(H') \, \mathcal{F}_M(H)^{+}, \tag{7}$$

where $^{+}$ is the matrix pseudoinverse. Similarly, in the controlled setting, the approximation of $\mathcal{K}$ separated into matrices $K$ and $B$,

$$[K, B] = \mathcal{F}_M(H') \, [\mathcal{F}_M(H), \mathcal{G}_{\hat{M}}(X)]^{+},$$

where $\mathcal{G}_{\hat{M}}$ is the second dictionary mapping from $\mathbb{R}^{d_x}$ to $\mathbb{R}^{\hat{M}}$. For more on the Koopman operator in the controlled setting, as well as EDMD, see Appendix A.2. Regardless of whether uncontrolled or controlled, the mapping $C$ projects down to the state space from the high dimensional dictionary space and is approximated by minimizing $\|H - C\mathcal{F}_M(H)\|$, hence

$$C = H\mathcal{F}_M(H)^{+}.$$

We connect the EDMD algorithm to our recurrent network in Equation (6) by choosing $\mathcal{F}_M(\boldsymbol{h}) = \sigma(W_h \, \boldsymbol{h} + \boldsymbol{b}_h)$ and $\mathcal{G}_{\hat{M}}(\boldsymbol{x}) = \sigma(W_x \, \boldsymbol{x} + \boldsymbol{b}_x)$. With this setting, we see that approximating $K$ and $B$ in Equation (6) can be seen as a Koopman approximation with the dictionary being a hidden layer with $M$ and $\hat{M}$ neurons respectively. This connection highlights the benefit of operating in a higher dimensional space. It also allows us to make use of Koopman theory in the next section as the EDMD approximation is known to converge to $\mathcal{K}$. Finally, it is important to notice that the resulting function in Equation (6) consists of two neural networks applied to the previous state and input. The hidden layers are sampled, and the outer matrices are constructed using linear solvers. Hence, Equation (6) is a sampled recurrent neural network. The methods above can be summarized by sampling weights in Algorithm 1, constructing the RNN in Algorithm 2, prediction for uncontrolled systems in Algorithm 3, and model predictive control in Algorithm 4.

---

**Algorithm 1** Sampling weights and bias for a given dataset and probability distribution.

> **procedure** SAMPLE-LAYER($Z, \mathbb{P}_{\mathcal{Z}}$)
>   $W_z \in \mathbb{R}^{M \times d_z}, \boldsymbol{b}_z \in \mathbb{R}^{d_z}$
>   **for** $j = 1, 2, \ldots M$ **do**
>     Sample $(\boldsymbol{z}^{(1)}, \boldsymbol{z}^{(2)}) \sim \mathbb{P}_z$ from sample space $Z \times Z$
>     $W_z^{[j,:]} = \frac{\boldsymbol{z}^{(2)} - \boldsymbol{z}^{(1)}}{\|\boldsymbol{z}^{(2)} - \boldsymbol{z}^{(1)}\|^2}^{\mathsf{T}}$
>     $\boldsymbol{b}_z^{[j]} = -\langle (W_z^{[j,:]})^{\mathsf{T}}, \boldsymbol{z}^{(1)} \rangle$
>   **end for**
>   Return $W_z, \boldsymbol{b}_z$
> **end procedure**

**Algorithm 2** Sampling RNNS for the controlled setting with output $\boldsymbol{y}$.

> **procedure** SAMPLE-RNN($X, Y, H, H'$)
>   $W_x, \boldsymbol{b}_x \leftarrow$ SAMPLE-Layer($X, \mathbb{P}_X$)
>   $W_h, \boldsymbol{b}_h \leftarrow$ SAMPLE-Layer($H, \mathbb{P}_H$)
>   $\mathcal{F}_M(\cdot), \mathcal{G}_{\hat{M}}(\cdot) \leftarrow \sigma(W_h \cdot + \boldsymbol{b}_h), \sigma(W_x \cdot + \boldsymbol{b}_x)$
>   $[K, B] = \mathcal{F}_M(H')[\mathcal{F}_M(H), \mathcal{G}_{\hat{M}}(X)]^{+}$
>   $C = H \, \mathcal{F}_M(H')\mathcal{F}_M(H)^{+}$
>   $V = YH^{+}$
>   Return $V, CK\mathcal{F}_M, CB\mathcal{G}_{\hat{M}}$
> **end procedure**

**Algorithm 3** Prediction of new trajectory uncontrolled inputs using sampled RNN.

---

**procedure** PREDICT($Y, H, H', \boldsymbol{h}_0$,T)
    $F_{\mathcal{Y}}, F_{\mathcal{H}} \leftarrow$ SAMPLE-RNN($0, Y, H, H'$)
    **for** $t = 1, 2, \ldots ,$T **do**
        $\boldsymbol{h}_t \leftarrow F_{\mathcal{H}}(\boldsymbol{h}_{t-1})$
        $\boldsymbol{y}_t = F_{\mathcal{Y}}(\boldsymbol{h}_t)$
    **end for**
    Return $\{\boldsymbol{h}_t, \boldsymbol{y}_t\}_{t=1}^T$
**end procedure**

**Algorithm 4** Model predictive control (MPC) of system $F$ using LQR and sampled RNN.

---

**procedure** MPC($X, Y, H, H', \boldsymbol{h}_0$,T,$\boldsymbol{h}^*$)
    $F_{\mathcal{Y}}, F_{\mathcal{X}}, F_{\mathcal{H}} \leftarrow$ SAMPLE-RNN($X, Y, H, H'$)
    lqr $\leftarrow$ LQR.fit($F_{\mathcal{X}}, F_{\mathcal{H}}$)
    lqr.set_target($\boldsymbol{h}^*$)
    **for** $t = 1, 2, \ldots ,$T **do**
        $\boldsymbol{x}_{t-1} \leftarrow$ lqr.control_sequence($\boldsymbol{h}_{t-1}$)
        $\boldsymbol{h}_t \leftarrow F(\boldsymbol{h}_{t-1}, \boldsymbol{x}_{t-1})$
        $\boldsymbol{y}_t = F_{\mathcal{Y}}(\boldsymbol{h}_t)$
    **end for**
    Return $\{\boldsymbol{x}_t, \boldsymbol{h}_t, \boldsymbol{y}_t\}_{t=1}^T$
**end procedure**

### 3.3 CONVERGENCE OF SAMPLED RNNS

Under some conditions, the convergence of sampled RNNs for uncontrolled systems can be shown for arbitrary finite horizon predictions. The result shows convergence by estimating $K$ using Equation (7). This differs from the usual existence proofs for parameters of RNNs, and is possible due to the Koopman connection established in the previous section.

With $L^2 := L^2(\mathcal{H}, \mu_h)$ being the usual Lebesgue space, we can state the required assumptions.

**Assumption 1.** The assumptions on $\mu_h$, $\mathcal{F}_M$, and $F$ are the following.

1. $\mu_h$ is regular and finite for compact subsets.

2. Hidden layer $\mathcal{F}_M$ must fulfill $\mu_h\{\boldsymbol{h} \in \mathcal{H} \mid \boldsymbol{c}^{\mathsf{T}}\mathcal{F}_M(\boldsymbol{h}) = 0\} = 0$, for all nonzero $\boldsymbol{c} \in \mathbb{R}^M$.

3. The Koopman operator $\mathcal{K}: L^2 \to L^2$ is a bounded operator.

The first two points are not very restrictive and hold for many measures and activation functions (such as $tanh$ activation function and the Lebesgue measure). The third assumption is common when showing convergence in Koopman approximation theory and holds for a broad set of dynamical systems (See Appendix B.1 for further discussion of all three points). Finally, we also require $\mathcal{H}$ to follow Definition 2 in Appendix B.1.

We now denote $L_K^2$ as the space of vector valued functions functions $f = [f_1, f_2, \ldots, f_K]$, where $f_i \in L^2$ and $\|f\| = \sum_{k=1}^K \|f_k\|_{L^2}$. We let $F^t(\boldsymbol{h}_0) = \boldsymbol{h}_t$ be the true state after time $t$, and $K_N$ be the solution of Equation (7), where $N$ data points have been used to solve the least square problem.

**Theorem 1.** *Let $f \in L_K^2$, $H, H'$ be the dataset with $N$ data points used in Equation* (7)*, and Assumption 1 holds. For any $\epsilon > 0$ and $T \in \mathbb{N}$, there exist an $M \in \mathbb{N}$ and hidden layers $\mathcal{F}_M$ and matrices $C$ such that*

$$\lim_{N \to \infty} \int_{\mathcal{H}} \|CK_N^t \mathcal{F}_M - f \circ F^t\|_2^2 d\mu_h < \epsilon,$$

*for all $t \in [1, 2, \ldots, T]$.*

For prediction of the system output, as the identity function $Id(\boldsymbol{h}) \mapsto \boldsymbol{h}$ is in $L_{d_h}^2$, the result above implies convergence of $\int_{\mathcal{H}} \|CK_N^t \mathcal{F}_M - F^t\|_2^2 d\mu_h$. The proof can be found in Appendix B.1, and in Appendix B.2 we discuss the limitations of the result w.r.t. the controlled setting.

## 4 COMPUTATIONAL EXPERIMENTS

We now discuss a series of experiments designed to illustrate the benefits and challenges of our construction approach. We compare our method to the state-of-the-art iterative gradient-based method called shPLRNN, which we explain in Appendix D. For the real-world weather dataset, we also compare our approach with a long short-term memory (LSTM) model. Furthermore, due to the similarities our method bears with reservoir models, we also compare with an established reservoir model, namely an echo state network (ESN), further explained in Appendix E. Details on the datasets can be found in Appendix F, hyperparameters for all models are given in Appendix H, the evaluation

Table 1: Results from computational experiments. We report the training time and MSE (mean squared error) or EKL (empirical Kullback–Leibler divergence, see Appendix G) for a sampled RNN (our approach), a reservoir model ESN (see Appendix E), a state of the art backpropagation-based RNN called shPLRNN (see Appendix D), and a long short-term memory (LSTM) model.

| Example | Model | Time [s]: avg (min, max) | MSE: avg (min, max) |
|---|---|---|---|
| Van der Pol | sampled RNN | 0.26 (0.18, 0.35) | 9.55e-4 (7.08e-4, 1.28e-3) |
| | ESN | 3.76 (2.29, 5.40) | 1.58e-2 (1.15e-2, 2.07e-2) |
| | shPLRNN | 217.93 (203.10, 251.51) | 1.39e-2 (5.66e-3, 3.00e-2) |
| 1D Van der Pol | sampled RNN | 0.29 (0.25, 0.31) | 5.06e-3 (1.57e-4, 1.58-2) |
| Weather (day) | sampled RNN | 4.87 (4.83, 4.92) | $2.239°C$ $(2.088°C, 2.392°C)$ |
| | LSTM | 378.10 (284.00, 421.30) | $2.531°C$ $(2.183°C, 2.754°C)$ |
| | shPLRNN | 321.76 (298.98, 384.46) | $2.296°C$ $(1.803°C, 2.548°C)$ |
| Weather (week) | sampled RNN | 4.87 (4.81, 4.90) | $4.624°C$ $(4.169°C, 4.867°C)$ |
| | LSTM | 628.10 (580.70, 648.50) | $4.544°C$ $(3.964°C, 4.893°C)$ |
| | shPLRNN | 830.84 (796.55, 847.32) | $2.604°C$ $(2.500°C, 2.801°C)$ |
| Example | Model | Time [s]: avg (min, max) | EKL: avg (min, max) |
| Lorenz-63 | sampled RNN | 1.67 (1.34, 1.92) | 4.36e-3 (3.66e-3, 5.36e-3) |
| | ESN | 3.54 (2.87, 4.47) | 8.73e-3 (7.20e-3, 1.06e-2) |
| | shPLRNN | 607.42 (581.39,650.56) | 5.79e-3(4.41e-3,7.56e-3) |
| Rössler | sampled RNN | 5.36 (4.39, 6.39) | 1.57e-4 (5.86e-5, 3.82e-4) |
| | ESN | 8.11 (7.94, 8.31) | 8.33e-5 (3.79e-5, 2.25e-4) |
| | shPLRNN | 866.17 (848.56, 939.06) | 6.53e-4 (4.35e-4,1.09e-3) |

metrics are explained in Appendix G and a further comparison discussion as well as the hardware details are provided in Appendix H.

In Table 1, we list the quantitative results of the experiments without control. Each entry stems from five different runs, where the random seed is changed in order to ensure a more robust result. We give the mean over these five runs, as well as the minimum and maximum among them.

## 4.1 SIMPLE ODES: VAN DER POL OSCILLATOR

We consider the Van der Pol oscillator system for a simple illustration of our method. A sampled RNN with a $tanh$ activation and a single hidden layer of width 80 is used, and the prediction method follows Algorithm 3. The model is evaluated on test data; the averaged error and training time are reported in Table 1. One trajectory from the test set is visualized in Figure 2. It should be noted that predictions start with an initial condition from the test dataset which is used to make the first prediction, and afterwards continue using this prediction as an input to predict the next state, without information from the ground-truth dataset. In the results we observe a very stable trajectory over a long prediction horizon, and furthermore all eigenvalues of the Koopman operator are inside the unit disk (see Appendix C), thus we are certain that the model is stable. This experiment is also significant due to the periodic nature of the system, which is captured with our model, although neural network architectures in general struggle to capture periodicity (Ziyin et al., 2020). Compared to the gradient-descent trained shPLRNN our method is much faster and achieves higher forecasting accuracy, with lower MSE as prediction error. Compared with an ESN, our model has a shorter fit time and a smaller error. However, the hyperparameter search is simpler for our method since there are fewer hyperparameters to tune.

## 4.2 EXAMPLE WITH TIME DELAY EMBEDDING: VAN DER POL OSCILLATOR

For many real-world examples, it is not possible to observe the full state of a system. Here, we use the same datasets as in the simple Van der Pol experiment (Section 4.1) but only consider the first coordinate $h_1$. We embed the data using a time-delay embedding of six followed by a principal

component analysis (PCA) projection which reduces the dimensionality to two. A sampled RNN with $tanh$ activation and a single hidden layer of width 80 is trained. Predicted trajectories from the initial test dataset state are shown in the bottom row of the right column of Figure 2. The fit time and MSE error are provided in Table 1, they are fairly similar to the fit time and error for the example where the full state is observed indicating that this model also captures the true dynamics.

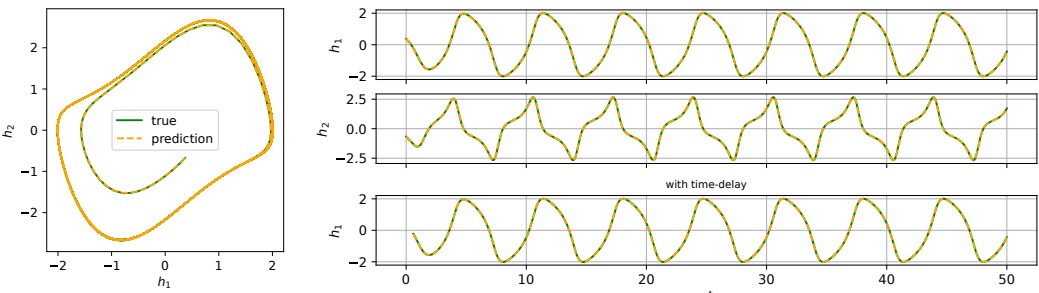

Figure 2: Comparison of true and predicted trajectories fror the Van der Pol experiments are shown for a test trajectory. Left: state space representation. Right: the top two rows show the full state system's first and second coordinate from Section 4.1, and the bottom most row shows the partially observed system from Section 4.2.

### 4.3 EXAMPLES OF CHAOTIC DYNAMICS: LORENZ AND RÖSSLER SYSTEMS

Chaotic systems pose a challenging forecasting problem from the class of dynamical systems. As an example, we consider the well-known Lorenz system in the chaotic regime. A sampling network with a $tanh$ activation and a single hidden layer of width 200 is trained. Predictions for a test trajectory are visualized in Figure 3. To evaluate the model, we do not calculate MSE since it is not suitable for chaotic trajectories, but instead an empirical KL divergence (EKL), to compare the orbits. For details on this geometric measure see Appendix G. The averaged EKL and training time are reported in Table 1. Our sampled RNN achieves a comparable performance with the reservoir model. However, it should be noted that training a reservoir model on Lorenz data is a well-studied problem, and the choice of hyperparameters has been tuned carefully to achieve excellent performance. When choosing hyperparameters for our sampling RNN we found that the necessary effort is low, as there are not as many degrees of freedom as in a reservoir. On the other hand, compared with the shPLRNN trained with gradient descent, we observe a much better performance both in terms of error and training time.

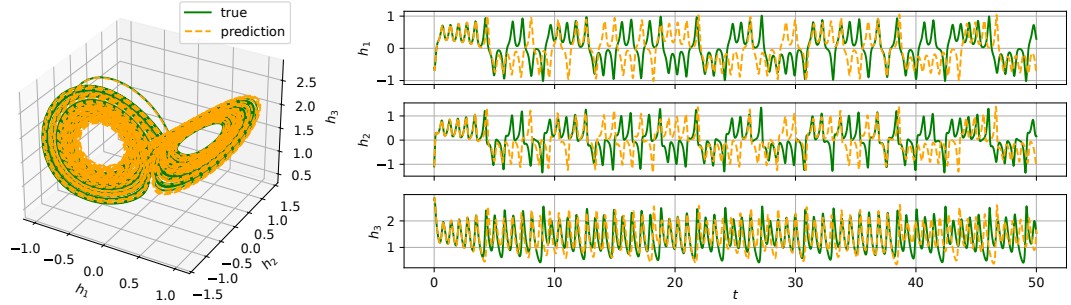

Figure 3: The results from the Lorenz experiment are shown for a test trajectory.
Left: state space representation of true and predicted trajectories. Right: trajectories obtained from the Lorenz model described in Section 4.3.

Furthermore, we consider the Rössler system in the chaotic regime. We use a sampled RNN with 300 hidden layer nodes and $tanh$ activation. A predicted test trajectory is shown in Figure 4. Since the system is chaotic, we calculate an EKL for our model and report it along with the training time in Table 1. Our model requires a slightly shorter fit time than the ESN and achieves comparable performance in terms of the EKL error. As depicted in Table 1, our method for chaotic systems is

significantly faster and more accurate at forecasting, with lower prediction error EKL, compared to the gradient-based model. Additionally, training with iterative methods requires many hyperparameters that need careful tuning to achieve optimal performance. Efficiently finding the best hyperparameters can be very challenging, especially when there is a small amount of training data for chaotic trajectories, making the training of such data more difficult. However, our training method performs effectively even with a very small amount of training data.

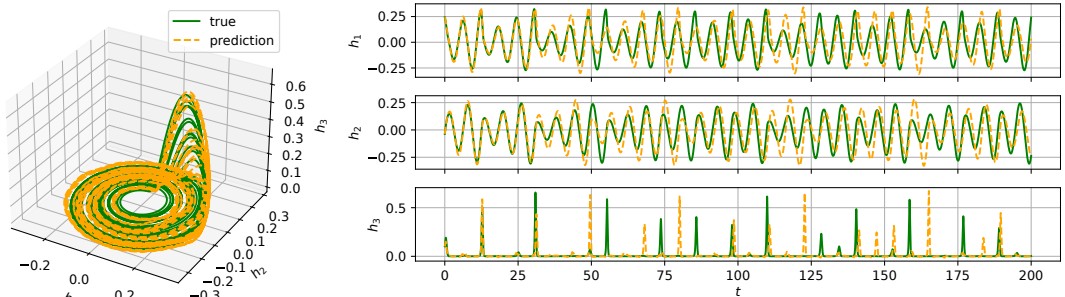

Figure 4: Trajectories from the Rössler experiment are shown for a test trajectory. Left: state space representation of true and predicted trajectories. Right: trajectories obtained from the Rössler model described in Section 4.3.

### 4.4 EXAMPLE WITH CONTROL INPUTS: FORCED VAN DER POL OSCILLATOR

We consider again the Van der Pol oscillator, where now the second coordinate, $h_2$, is controlled with an external input $x$, and using a sampled RNN model we perform model predictive control (MPC) as in Algorithm 4. We let $\mathcal{G}_{\hat{M}}$ be the identity function, and let $B$ map from $\mathcal{X}$ to $\mathbb{R}^M$ (adding nonlinearity for $x$ did not yield different results for this system, and is described in Appendix H.1.1). The sampled recurrent neural network is then the identity $\mathcal{G}_{\hat{M}}$ and $\mathcal{F}_M$ is a hidden layer of width 128 and $tanh$ activation. This network is then passed as a surrogate model to a linear-quadratic regulator (LQR). The network, in combination with the LQR, can successfully steer the state to the target state (see Figure 5). We consider five different runs, where only the random seed is varied, and obtain the mean controller cost to be 125.92 and the mean training time of 1.122 seconds. The norm of the state is also tracked over time, for five different runs we show the norms and the pointwise mean (over the runs) in Figure 5. This experiment highlights a key advantage of our model, which allows for modelling a nonlinear system such as the Van der Pol oscillator using a linear controller such as LQR. This implies that the well-established tools from linear control theory can be applied to non-linear systems using our method.

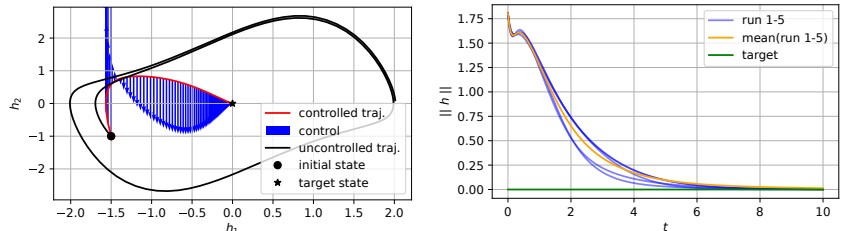

Figure 5: Controlled (i.e. forced) Van der Pol experiment (Section 4.4) for initial condition $\boldsymbol{h}_0 = [-1.5, -1]^\mathsf{T}$. Left: state space representation of controlled and uncontrolled trajectories. Right: $L^2$ norm of the controlled trajectory for five different runs and the $L^2$ norm of the target state.

### 4.5 EXAMPLE WITH REAL-WORLD DATA

**Weather data**   We apply our approach to the climate data presented in TensorFlow (2024). The dataset contains a time series of 14 weather parameters recorded in Jena (Germany) between January

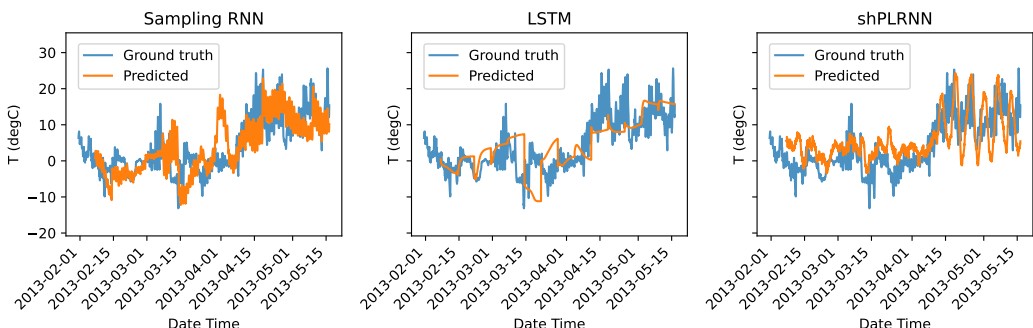

Figure 6: The predictions of the best models on the test dataset for the horizon of one week: sampled RNN (left), LSTM (middle), and shPLRNN (right).

1st, 2009, and December 31st, 2016. The data offers freedom in choosing the sizes of the time delay and prediction horizons. We decided to fix the time delay to one week and set two separate experiments with a prediction horizon of one day and one week. In the sampled RNN and LSTM experiments, we performed a grid search for each model with hyperparameters specified in Appendix H.4.1.

Table 1 shows the averaged training time and error metrics for the two selected horizons (day and week). We observe that the models perform similarly in the case of a shorter horizon, while sampling offers much faster training. When considering a one-week horizon, shPLRNN outperforms, while sampled RNN is still orders of magnitude faster. We also note the prediction horizon does not influence the training time of the sampled model that agrees with the Algorithm 2. When comparing predictions for the longer horizon in Figure 6, we notice that the LSTM struggles to predict the high-frequency fluctuations of the measurement, but the sampled RNN and shPLRNN successfully capture them. Figure 6 also highlights the deficiency of the MSE metric because a low mean error does not always correspond to accurate predictions, as illustrated by all three models. Overall, we conclude that sampled RNNs can successfully capture chaotic real-world dynamics and produce results comparable to the iterative models while offering a significant speed-up in training.

## 5    CONCLUSION

We introduce an efficient and interpretable training method for recurrent neural networks by combining ideas from random feature networks and Koopman operator theory.

**Benefits of the approach** In many examples, we demonstrate that we can train accurate recurrent networks orders of magnitude faster compared to networks trained by iterative methods. We also observe that the training approach works with a very small amount of training data. The direct connection to Koopman operator theory allows us to draw on existing theoretical results for dynamical systems, which we use to prove convergence in the limit of infinite width.

**Limitations compared to other methods** The training method we use involves the solution of a large, linear system. The complexity of solving this system depends cubically on the minimum number of neurons and the number of data points (respectively, time steps). This means if both the network and the number of data points grow together, the computational time and memory demands for training grow too quickly. With backpropagation-in-time, the memory requirements are mostly because many gradients must be stored for one update pass.

**Remaining challenges and future work** Recurrent networks are often used for tasks in computer vision and natural language processing. These tasks require network architectures beyond feed-forward networks, like convolutional neural networks or transformers, which means we currently struggle with high dimensional data. The sampling scheme we use to construct the hidden weights of the neurons is currently not useful in constructing parameters for such architectures, but it is an intriguing challenge to work towards sampling them. Remaining challenges in the theoretical work include extending the theory shown in this paper to controlled systems, as well as bridging the Koopman theory for continuous dynamical systems and NeuralODEs.

## 6 ETHICS STATEMENT

For comparatively small problems, our construction method is orders of magnitude faster to construct the parameters of a recurrent network. As neural networks are generally dual-use, our work potentially also allows faster training for misuse of this technology. Still, we connect the construction of recurrent neural networks to Koopman operator theory and dynamical systems. This connection allows researchers in these fields to better understand the behavior, failure modes, and robustness of recurrent architectures. In addition, by sampling weights and bias from the input spaces it adds interpretability to the models which again lets users understand better the underlying networks used (see Appendix C for an example of this). We believe that this far outweighs the potential downsides of misuse because recurrent architectures that are understood much better can also be regulated in a more straightforward way.

## 7 REPRODUCIBILITY STATEMENT

In our work we try hard to ensure that our results are robust and reproducible. In terms of theoretical statements, the assumptions are stated in the main paper and further discussed in the appendix. The complete detailed proofs can also be found in the Appendix B. In addition, we have added an overview of Koopman theory before the proofs to aid the understanding in Appendix A. When theoretical statements rely on other work, this is also clearly cited. For the computational experiments, we submit an anonymized code folder, and provide all hyperparameters in Appendix H. The code will also be open sourced upon acceptance. Furthermore, all reported results that are based on five runs with different random seeds, to ensure robustness.

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

## APPENDIX

## A KOOPMAN OPERATOR AND EXTENDED DYNAMIC MODE DECOMPOSITION

In this section we give a more thorough introduction to the Koopman operator and its use in dynamical system theory. In addition, we also walk the reader through extended dynamic mode decomposition (EDMD), which is the finite approximation of the Koopman operator we use in the main paper.

Consider a dynamical system $(\mathcal{H}, F)$, where the state space $\mathcal{H}$ is a topological space and $F\colon \mathcal{H} \to \mathcal{H}$ is an evolution operator/map. We impose more structure on our system by requiring our state space $\mathcal{H}$ to be a measure space $(\mathcal{H}, \mathcal{F}, \mu)$, and $F$ to be $\mathcal{F}$-measurable. We start by considering $(\mathcal{H}, F)$ to be a discrete-time system, which are the systems we mainly work with in this paper. We can then write the evolution as

$$\boldsymbol{h}_{t+1} = f(\boldsymbol{h}_t), \quad \boldsymbol{h}_t \in \mathcal{H} \subseteq \mathbb{R}^{d_h}, \quad t \geq 0. \tag{8}$$

The analysis of the evolution in the original state space can be hard, especially when $F$ is non-linear. The Koopman theory takes a different approach, and looks at the evolution of observables (e.g. measurements of the states) instead of the states themselves. The observables one is working with are typically $\phi\colon \mathcal{H} \to \mathbb{C}$ in a suitable Hilbert space $\mathcal{H}$. The evolution of the observables is then captured by the *Koopman operator* $\mathcal{K}$,

$$[\mathcal{K}\,\phi](\boldsymbol{h}) := (\phi \circ F)(\boldsymbol{h}).$$

As long as the space of observables is a vector space, the Koopman operator is linear and we may analyse the dynamical system with non-linear $F$ using spectral analysis, with the caveat that in the majority of cases $\mathcal{K}$ is infinite dimensional. The choice of the function space $\mathcal{F}$ is crucial, as $\phi \circ F$ must belong to $\mathcal{F}$ for all $\phi \in \mathcal{F}$. Assuming $F$ is measure-preserving — which is common in ergodic theory — one can address the issue by setting

$$\mathcal{F} = L^2(\mathcal{H}, \mu_h) \coloneqq \left\{ \phi\colon \mathcal{H} \to \mathbb{F} \;\middle|\; \|\phi\|_{L^2(\mathcal{H}, \mu_h)} = \left( \int_{\mathcal{H}} |\phi(\boldsymbol{h})|^2 \, \mu_h(d\boldsymbol{h}) \right)^{\frac{1}{2}} < \infty \right\}. \tag{9}$$

where $\mathbb{F} = \mathbb{R}$ or $\mathbb{C}$. As we consider spectral analysis in this section, we let $\mathbb{F} = \mathbb{C}$. Since $F$ is measure-preserving means $\mathcal{K}$ is an isometry and the issue is resolved. The map $\mathcal{K}$ might still not be well-defined, as $\phi_1, \phi_2 \in L^2(\mathcal{H}, \mu_h)$ may differ only on a null set, yet their images under $\mathcal{K}$, could differ over a set of positive measure. To exclude this possibility, $F$ must be $\mu_h$-nonsingular, meaning that for each $H \subseteq \mathcal{H}$, $\mu_h(F^{-1}(H)) = 0$ if $\mu_h(H) = 0$.

Once the function space $\mathcal{H}$ is chosen, making sure that $\mathcal{K}$ is well-defined, we may apply spectral analysis. A Koopman eigenfunction $\varphi_k \in L^2(\mathcal{H}, \mu_h)$ corresponding to a Koopman eigenvalue $\lambda_k \in \sigma(\mathcal{K})$ satisfies

$$\varphi_k(\boldsymbol{h}_{t+1}) = \mathcal{K}\varphi_\lambda(\boldsymbol{h}_t) = \lambda_k\,\varphi_\lambda(\boldsymbol{h}_t).$$

When the state space $\mathcal{H} \subseteq \mathbb{R}^d$, under certain assumptions of the space of eigenfunctions, we can evolve $\boldsymbol{h}$ using the spectrum of $\mathcal{K}$. More concretely, let $\Phi\colon \mathcal{H} \to \mathbb{C}^d$ be a vector of observables, where each observable $\phi_i(\boldsymbol{h}) = \boldsymbol{h}_i$, where $\boldsymbol{h}_i$ is the $i$th component of $\boldsymbol{h}$. Assuming $\phi_i \in \mathrm{Span}\{\varphi_k\} \subset L^2(\mathcal{H}, \mu_h)$, we have

$$\mathcal{K}\,\phi_i = \mathcal{K} \sum_k c_k\,\varphi_k = \sum_k c_k\,\mathcal{K}\,\varphi_k = \sum_k c_k\,\lambda_k\,\varphi_k,$$

due to the linearity of $\mathcal{K}$. Iterative mapping of $\mathcal{K}$ yields

$$\boldsymbol{h}_t = \Phi(\boldsymbol{h}_t) = (\underbrace{[\mathcal{K}\phi] \circ \cdots \circ [\mathcal{K}\phi]}_{t})(\boldsymbol{h}_0) = \sum_k \lambda_k^t\,\phi_{\lambda_k}(\boldsymbol{h}_0)\,\boldsymbol{c}_k^{\Phi}. \tag{10}$$

This process is known as *Koopman mode decomposition* (KMD), and the vectors $\boldsymbol{c}_k^{\Phi} \in \mathbb{C}^d$ are referred to as Koopman modes associated with the observable $\Phi$. This reveals one of the true strengths of the Koopman theory.

Up until now we have considered a discrete dynamical system, but the Koopman theory can also be extended to continuous systems where $\boldsymbol{h}$ is a function of time $t$ and its evolution is given by

$$\dot{\boldsymbol{h}}(t) = F(\boldsymbol{h}(t)), \quad \boldsymbol{h}(t) \in \mathcal{H}, t \in \mathbb{R}_{\geq 0},$$

by using the flow map. For any $t$, the flow map operator denoted by $F^t \colon \mathcal{H} \to \mathcal{H}$ is defined as

$$\boldsymbol{h}(t) = F^t(\boldsymbol{h}) = \boldsymbol{h}(0) + \int_0^t F(\boldsymbol{h}(\tau)) d\tau,$$

which maps from an initial condition $\boldsymbol{h}(0)$ to point on the trajectory $t$ time units away. We can then define the Koopman operator for each $t \in \mathbb{R}_{\geq 0}$ ,

$$[\mathcal{K}^t \phi](\boldsymbol{h}) = (\phi \circ F^t)(\boldsymbol{h}),$$

where $\phi \in L^2(\mathcal{H}, \mu_h)$. The set of all these operators $\{\mathcal{K}^t\}_{t \in \mathbb{R}_{\geq 0}}$ forms a semigroup with an infinitesimal generator $\mathcal{L}$. With some assumption on the continuity of the semigroup, the generator is the Lie derivative of $\phi$ along the vector field $F(\boldsymbol{h})$ and can be written as

$$[\mathcal{L}\phi](\boldsymbol{h}) = \lim_{t \downarrow 0} \frac{[\mathcal{K}^t \phi](\boldsymbol{h}) - \phi(\boldsymbol{h})}{t} = \frac{d}{dt}\phi(\boldsymbol{h}(t))\Big|_{t=0} = \nabla\phi \cdot \dot{\boldsymbol{h}}(0) = \nabla\phi \cdot F(\boldsymbol{h}(0)).$$

The eigenfunction and eigenvalue is in the continuous time case scalars and functions fulfilling

$$[\mathcal{K}^t \varphi_k](\boldsymbol{h}) = e^{\lambda_k t} \varphi_k(\boldsymbol{h}),$$

where $\{e^{\lambda_k}\}$ are the eigenvalues of the semigroup. This allows us to use the Koopman theory for continuous dynamical systems as well, and possibly make the connection for NeuralODEs and Koopman theory in similar fashion we have done with discrete system Koopman operator and RNN.

As the Koopman operator is infinite dimensional makes it impossible to apply it directly, and raises the need for a method to create a finite approximation of $\mathcal{K}$ and its spectrum, namely the extended dynamic mode decomposition.

## A.1 EXTENDED DYNAMIC MODE DECOMPOSITION

As we are mostly working with discrete systems in this paper, we focus on approximating the Koopman operator $\mathcal{K}$ for discrete dynamical systems. The way to approximate $\mathcal{K}$ is by extended dynamic mode decomposition (EDMD), which is an algorithm that provides a data driven finite dimensional approximation of the Koopman operator $\mathcal{K}$ through a linear map $K$. The spectral properties of $K$ subsequently serve to approximate those of $\mathcal{K}$. Utilizing this approach enables us to derive the Koopman eigenvalues, eigenfunctions, and modes. Here, we provide a brief overview of EDMD. For further details, refer to Williams et al. (2015b).

The core concept of EDMD involves approximating the operator's action on $\mathcal{F} = L^2(\mathcal{H}, \mu_h)$ by selecting a finite dimensional subspace $\widetilde{\mathcal{F}}_M \subset \mathcal{F}$. To define this subspace, we start by choosing a dictionary $\mathcal{F}_M = \{\psi_i \colon \mathcal{H} \to \mathbb{R} \in \mathcal{H} \mid i = 1, \cdots, M\}$. We the have

$$\mathcal{F}_M(\boldsymbol{h}) = [\psi_1(\boldsymbol{h}) \quad \psi_2(\boldsymbol{h}) \quad \cdots \quad \psi_M(\boldsymbol{h})]^\mathsf{T},$$

and we let the finite dimensional subspace $\widetilde{\mathcal{F}}_M$ be

$$\widetilde{\mathcal{F}}_M = \mathrm{Span}\{\psi_1, \psi_2, \cdots, \psi_M\} = \{\boldsymbol{a}^\mathsf{T} \mathcal{F}_M \colon \boldsymbol{a} \in \mathbb{C}^M\} \subset \mathcal{F}.$$

The action of the Koopman operator on $\phi \in \widetilde{\mathcal{F}}_M$ due to linearity is

$$\mathcal{K}\phi = \boldsymbol{a}^\mathsf{T} \mathcal{K} \mathcal{F}_M = \boldsymbol{a}^\mathsf{T} \mathcal{F}_M \circ F.$$

Assuming that the subspace $\widetilde{\mathcal{F}}_M$ is invariant under $\mathcal{K}$, i.e., $\mathcal{K}(\widetilde{\mathcal{F}}_M) \subseteq \widetilde{\mathcal{F}}$, we can write $\mathcal{K}\phi = \boldsymbol{b}^\mathsf{T} \mathcal{F}_M$ for any $\phi \in \widetilde{\mathcal{F}}_M$. It follows that $\mathcal{K}|_{\widetilde{\mathcal{F}}_M}$ is finite dimensional and can be written as a matrix $K \in \mathbb{R}^{M \times M}$ such that $\boldsymbol{b}^\mathsf{T} = \boldsymbol{a}^\mathsf{T} K$. The equality can be seen, where $\phi \in \widetilde{\mathcal{F}}_M$, through

$$K\phi = \boldsymbol{a}^\mathsf{T} K \mathcal{F}_M = \boldsymbol{b}^\mathsf{T} \mathcal{F}_M = \mathcal{K}\phi$$

where one first applies linearity of $K$, then the invariant assumption. When $\widetilde{\mathcal{F}}_M$ is not an invariant subspace of the Koopman operator $\mathcal{K}$, $K$ becomes an approximation. Decomposing $\mathcal{K}\phi = \boldsymbol{b}^\mathsf{T}\mathcal{F}_M + \rho$, where $\rho \in L^2(\mathcal{H}, \mu_h)$, EDMD approximates $\mathcal{K}$ by using data

$$H = [\boldsymbol{h}_1 \quad \boldsymbol{h}_2 \quad \cdots \quad \boldsymbol{h}_N], \quad H' = [\boldsymbol{h}'_1 \quad \boldsymbol{h}'_2 \quad \cdots \quad \boldsymbol{h}'_N],$$

where $\boldsymbol{h}'_n = F(\boldsymbol{h}_n)$. Then, to find $K$, the EDMD minimizes the following cost function

$$\mathcal{J} = \frac{1}{2}\sum_{n=1}^{N}\|\rho(\boldsymbol{h}_n)\|^2 = \frac{1}{2}\sum_{n=1}^{N}\left\|\boldsymbol{a}^\mathsf{T}\left(\mathcal{F}_M(\boldsymbol{h}'_n) - K\mathcal{F}_M(\boldsymbol{h}_n)\right)\right\|^2 \tag{11}$$

Letting

$$\mathcal{F}_M(H) = [\mathcal{F}_M(\boldsymbol{h}_1), \mathcal{F}_M(\boldsymbol{h}_2)\ldots, \mathcal{F}_M(\boldsymbol{h}_N)] \in \mathbb{C}^{K \times N}$$

and equivalently for $\mathcal{F}_M(H')$, a solution to Equation (11) is given by

$$K = \mathcal{F}_M(H')\,\mathcal{F}_M(H)^+ \tag{12}$$

where $\mathcal{F}_M(H)^+$ denotes the pseudo-inverse.

Upon obtaining $K$, we find approximations of eigenfunctions

$$\varphi_k = \xi_k \mathcal{F}_M,$$

where $\lambda_k$ and $\xi_k$ are eigenvalue and left eigenvector of $K$ respectively. Finally, denoting $\boldsymbol{\varphi}\colon \mathcal{H} \to \mathbb{C}^K$ as the function $\boldsymbol{h} \mapsto [\varphi_1(\boldsymbol{h}), \varphi_2(\boldsymbol{h}), \ldots, \varphi_K(\boldsymbol{h})]$, we approximate the Koopman modes by

$$C = \underset{\tilde{C} \in \mathbb{C}^{d \times K}}{\arg\min}\|\Phi(H) - \tilde{C}\boldsymbol{\varphi}(H)\|^2_{Fr} = \underset{\tilde{C} \in \mathbb{C}^{d \times K}}{\arg\min}\|H - \tilde{C}\boldsymbol{\varphi}(H)\|^2_{Fr},$$

where $\|\cdot\|_{Fr}$ is the Frobenius norm and $\Phi(\boldsymbol{h}) = \boldsymbol{h}$, as defined in previous section. We may now approximate Equation (10) as

$$\boldsymbol{h}_t = \Phi(\boldsymbol{h}_t) = C\,\Lambda^t\boldsymbol{\varphi}(\boldsymbol{h}_0),$$

where $\Lambda$ is a diagonal matrix with the eigenvalues. Due to numerical issues, one typically predict one time step at the time, and project down to state space each time.

In many applications, one is not necessarily interested in the spectral analysis, but only prediction. One then typically set $\mathbb{F} = \mathbb{R}$, solve for $K$ in exact way, but solve for $C$ as

$$C = \underset{\tilde{C} \in \mathbb{R}^{d \times K}}{\arg\min}\|\Phi(H) - \tilde{C}\mathcal{F}_M(H)\|^2_{Fr} = \underset{\tilde{C}}{\arg\min}\|H - \tilde{C}\mathcal{F}_M(H)\|^2_{Fr}.$$

For prediction one simply apply $K$ several times,

$$\boldsymbol{h}_t = CK^t\mathcal{F}_M(\boldsymbol{h}_0).$$

In practice, one usually predict step by step $\boldsymbol{h}_t$, that is, maps it down to the state space and maps back to the observable image space, before applying $K$ again. It is also worth noting that one may be more interested in mapping to an output of a function $f \in \mathcal{F}$ instead, and then one simply swap $\Phi$ with $f$ when approximating $C$.

To conclude this introduction to EDMD, we do want to mention that the set of eigenfunctions we find through the EDMD algorithm comes with its own set of issues, such as spectral pollution, and efforts to mitigate certain issues has spawned extensions to EDMD, e.g., measure-preserving EDMD and residual DMD (Colbrook, 2022; Colbrook et al., 2023).

## A.2 Controlled dynamical systems and the Koopman operator

Extending Koopman theory to controlled systems can be done in several ways, and we opt to follow Korda & Mezić (2018a) and limit ourselves to linear controlled systems,

$$\boldsymbol{h}_t = F(\boldsymbol{h}_{t-1}, \boldsymbol{x}_t) = A_h\boldsymbol{h}_{t-1} + A_x\boldsymbol{x}_t, \quad \boldsymbol{y}_t = A_y\boldsymbol{h}_t. \tag{13}$$

Rewriting the input and the evolution operator slightly allows us to apply the Koopman operator and its theory as described in previous sections. Let

$$\tilde{\boldsymbol{h}} = \begin{bmatrix} \boldsymbol{h} \\ \tilde{\boldsymbol{x}} \end{bmatrix},$$

where $\boldsymbol{h} \in \mathcal{H}$ and $\tilde{\boldsymbol{x}} \in \ell(\mathcal{X})$, with $\ell(\mathcal{X})$ is the space of all countable sequences $\{\boldsymbol{x}_i\}_{i=1}^{\infty}$ such that $\boldsymbol{x}_i \in \mathcal{X}$. Then we can rewrite the evolution operator $F \colon \mathcal{H} \times \mathcal{X} \to \mathcal{H}$ to $\tilde{F} \colon \mathcal{H} \times \ell(\mathcal{X}) \to \mathcal{H}$, where

$$\tilde{\boldsymbol{h}}_t = \tilde{F}(\tilde{\boldsymbol{h}}_{t-1}) = \begin{bmatrix} F(\boldsymbol{h}_{t-1}, \tilde{\boldsymbol{x}}(0)) \\ \mathcal{S}\,\tilde{\boldsymbol{x}} \end{bmatrix},$$

where $\tilde{\boldsymbol{x}}(i) = \boldsymbol{x}_i \in \tilde{\boldsymbol{x}}$ and $\mathcal{S}$ is the left-shift operator, i.e., $\mathcal{S}(\tilde{\boldsymbol{x}}(i)) = \tilde{\boldsymbol{x}}(i+1)$. The Koopman operator $\mathcal{K}$ can be applied to $\tilde{F}$ with observables $\phi \colon \mathcal{H} \times \ell(\mathcal{X}) \to \mathbb{C}$, and the rest of the Koopman theory follows.

When approximating the Koopman operator for controlled systems with EDMD, the dictionary we choose needs alteration due to the domain $\mathcal{H} \times \ell(\mathcal{X})$ is infinite dimensional. Korda & Mezić (2018a) proposes dictionaries that are both computable and enforces the linearity relationship assumed in Equation (13). The dictionaries to be considered are on the form

$$\psi_i(\boldsymbol{h}, \tilde{\boldsymbol{x}}) = \psi_i^{(h)}(\boldsymbol{h}) + \psi_i^{(x)}(\tilde{\boldsymbol{x}}),$$

where $\psi_i^{(x)} \colon \ell(\mathcal{X}) \to \mathbb{R}$ is a linear functional and $\psi_i^{(h)} \in \mathcal{F}$. The new dictionary can be written as

$$\mathcal{F}_M = \{\psi_1^{(h)}, \ldots, \psi_M^{(h)}\}, \quad \mathcal{G}_{\tilde{M}} = \{\psi_1^{(x)}, \ldots, \psi_{\tilde{M}}^{(x)}\}.$$

Note the number of observables $M$ and $\tilde{M}$ can differ, even though in the main paper we typically set $\tilde{M} = M$. If they differ, the matrix $B$ will map from $\mathbb{F}^{\tilde{M}}$ to $\mathbb{F}^M$. Once the dictionaries are set, we simply solve the optimization problem

$$\underset{K \in \mathbb{F}^{M \times M}, B \in \mathbb{F}^{M \times \tilde{M}}}{\arg\min} \frac{1}{2} \sum_{n=1}^{N} \|\mathcal{F}_M(\boldsymbol{h}_n') - (K\mathcal{F}_M(\boldsymbol{h}_n) + B\mathcal{G}_{\tilde{M}}(\boldsymbol{h}_n))\|^2$$

with the analytical solution being

$$[K, B] = \mathcal{F}_M(H')[\mathcal{F}_M(H), \mathcal{G}_{\tilde{M}}(H)]^+.$$

Approximating $C$ is done in the same manner as for uncontrolled systems, as it only needs to learn how to map from $\mathcal{F}_M(\mathcal{H})$ to $\mathcal{H}$. For further details, see Korda & Mezić (2018a).

## B  THEORY

In this section we give the necessary assumptions and proofs for the theoretical results in the main paper. We start by defining the state space $\mathcal{H}$ and input space $\mathcal{X}$, following the setup from Bolager et al. (2023). Letting

$$d_{\mathbb{R}^{d_z}}(\boldsymbol{z}, A) = \inf\{d(\boldsymbol{z}, \boldsymbol{a}) \colon \boldsymbol{a} \in A\}.$$

where $d$ is the canonical Euclidean distance in the space $\mathbb{R}^{d_z}$. The medial axis is defined as

$$\text{Med}(A) = \{\boldsymbol{h} \in \mathbb{R}^{d_z} \colon \exists \boldsymbol{p} \neq \boldsymbol{q} \in A, \|\boldsymbol{p} - \boldsymbol{z}\| = \|\boldsymbol{q} - \boldsymbol{z}\| = d_{\mathbb{R}^{d_z}}(\boldsymbol{z}, A)\}$$

and the reach is the scalar

$$\tau_A = \inf_{\boldsymbol{a} \in A} d_{\mathbb{R}^{d_z}}(\boldsymbol{a}, \text{Med}(A)),$$

i.e., the point in $A$ that is closest to the projection of points in $A^c$.

**Definition 2.** *Let $\widetilde{\mathcal{H}}$ be a nonempty compact subset of $\mathbb{R}^{d_h}$ with reach $\tau_{\widetilde{\mathcal{H}}} > 0$ and equivalently for $\widetilde{\mathcal{X}} \in \mathbb{R}^{d_x}$. The input space $\mathcal{H}$ is defined as*

$$\mathcal{H} = \{\boldsymbol{h} \in \mathbb{R}^{d_h} \colon d_{\mathbb{R}^{d_h}}(\boldsymbol{h}, \widetilde{\mathcal{H}}) \leq \epsilon_{\mathcal{H}}\},$$

*where $0 < \epsilon_{\mathcal{H}} < \min\{\tau_{\widetilde{\mathcal{H}}}, 1\}$. Equivalently for $\mathcal{X}$,*

$$\mathcal{X} = \{\boldsymbol{x} \in \mathbb{R}^{d_x} \colon d_{\mathbb{R}^{d_x}}(\boldsymbol{x}, \widetilde{\mathcal{X}}) \leq \epsilon_{\mathcal{X}}\},$$

*where $0 < \epsilon_{\mathcal{X}} < \min\{\tau_{\widetilde{\mathcal{X}}}, 1\}$.*

*Remark* 2. This restriction to the type of state and input spaces we consider is sufficient to construct all neural networks of interest by choosing pair of points from the space in question, and construct the weight and bias as in Equation (4). It is also argued in Bolager et al. (2023) that most interesting real-world application will contain some noise and make sure that the state and input spaces is approximately on the form given in the definition.

As we are not considering the eigenfunctions of the system, only prediction, and we are working with real valued neural networks, we are in the setting with $\mathbb{F} = \mathbb{R}$ in Equation (9).

### B.1 Uncontrolled systems

For uncontrolled systems we have that the evolution is described as $\boldsymbol{h}_t = F(\boldsymbol{h}_{t-1})$. For the following theory, we have the following assumptions.

**Assumption 2.** The measure defining the space $\mathcal{F} = L^2(\mathcal{H}, \mu_h)$, we assume $\mu_h$ is regular and finite for compact subsets.

**Assumption 3.** The following assumptions is made for the dictionary and the underlying system $F$:

1. Any dictionary $\mathcal{F}_M$ must fulfill $\mu_h\{\boldsymbol{h} \in \mathcal{H} \mid \boldsymbol{c}^\mathsf{T}\mathcal{F}_M(\boldsymbol{h}) = 0\} = 0$, for all nonzero $\boldsymbol{c} \in \mathbb{R}^M$.

2. $\mathcal{K}\colon \mathcal{F} \to \mathcal{F}$ is a bounded operator.

Assumption 2 is not very limited, as it holds for most measures we are interested in, such as measures absolutely continuous to the Lebesgue measure. For Assumption 3.1 we have the following result.

**Lemma 1.** *Let* $(\mathbb{R}^{d_h}, \mathscr{B}(\mathbb{R}^{d_h}), \mu_h)$ *be a measurable space with* $supp(\mu_h) = \mathcal{H}$*, and* $\lambda$ *be the Lebesgue measure for* $\mathbb{R}^{d_h}$*. If for all non-zero* $\boldsymbol{c} \in \mathbb{R}^{d_h}$*, the following holds:*

- *The set of functions* $\{\psi_1, \ldots, \psi_M\}$ *is linear independent.*

- $\boldsymbol{c}^\mathsf{T}\mathcal{F}_M = \sum_{m=1}^M c_m\psi_m$ *is analytic on* $\mathbb{R}^{d_h}$,

- $\mu_h \ll \lambda$,

*then Assumption 3.1 holds. In particular, it holds when* $\{\psi_i\}_{m=1}^M$ *are independent tanh functions.*

*Proof.* Let $\boldsymbol{c} \in \mathbb{R}^{d_h}$ be any non-zero vector. As $\{\psi_1, \ldots, \psi_M\}$ are linear independent, we have $\boldsymbol{c}^\mathsf{T}\mathcal{F}_M \not\equiv 0$. As $\boldsymbol{c}^\mathsf{T}\mathcal{F}_M$ is analytic, we have the set

$$A = \{\boldsymbol{h} \in \mathbb{R}^{d_h} \mid \boldsymbol{c}^\mathsf{T}\mathcal{F}_M(\boldsymbol{h}) = 0\},$$

to have measure zero, $\lambda(A) = 0$, due to Proposition 1 in Mityagin (2020). We have $\lambda(A \cap \mathcal{H}) \leq \lambda(A) = 0$. Finally due to absolutely continuous measure $\mu_h$, we have

$$\mu_h(\{\boldsymbol{h} \in \mathcal{H}\colon \boldsymbol{c}^\mathsf{T}\mathcal{F}_M(\boldsymbol{h}) = 0\}) = \lambda(A \cap \mathcal{H}) = 0,$$

and hence Assumption 3.1 holds. As the linear projection and shift of bias is analytic on $\mathbb{R}^{d_h}$, tanh is analytic on $\mathbb{R}$, and analytic functions are closed under compositions, means Assumption 3.1 holds when $\mathcal{F}_M$ is a set of linearly independent neurons with tanh activation function. $\square$

*Remark* 3. The requirement of the functions being analytic for the whole $\mathbb{R}^{d_h}$ can certainly be relaxed if necessary to an open and connected set $U$, s.t. $\mathcal{H} \subseteq U$). Further relaxation can be made with some additional work. The result above also agrees with the claim made in Korda & Mezić (2018b) about Assumption 3.1 holds for many measures and most basis functions such as polynomials and radial basis functions. Finally, we note that the independence requirement is easily true when we sample the neurons.

Assumption 3.2 is commonly enforced in Koopman theory when considering convergence of EDMD and for example holds when $F$ is Lipschitz, has Lipschitz invertible, and $\mu_h$ is the Lebesgue measure (Korda & Mezić, 2018b).

We note

$$\mathcal{NN}_{[1,1:\infty]} = \bigcup_{M=1}^\infty \mathcal{NN}_1(\mathcal{H}, \mathbb{R}^M)$$

is the space of all hidden layers with tanh activation function from $\mathcal{H}$. We continue with a result relating this space with $\mathcal{F}$.

**Lemma 2.** *For tanh activation function and when Assumption 2 holds, then $\mathcal{NN}_{[1,1:\infty]}$ is dense in $\mathcal{F}$ and has a countable basis $\{\psi_i \in \mathcal{NN}_{[1,1:\infty]}\}_{i=1}^{\infty}$, both when the parameter space is the full Euclidean space and when constructed as in Equation (4).*

*Proof.* It is well known that such space is dense in $C(\mathcal{H}, \mathbb{R}^K)$ for any $K \in \mathbb{N}$. This holds both when the weight space is the full Euclidean space (Cybenko, 1989; Pinkus, 1999) and when limited to the weight construction in Equation (4) (Bolager et al., 2023). As $\mathcal{H}$ is compact, we have that $\mathcal{NN}_{[1,1:\infty]}$ is dense in $\mathcal{F}$. Furthermore, as $\mathcal{F}$ is a separable Hilbert space and metric space, there exists a countable subset $\{\psi_i \in \mathcal{NN}_{[1,1:\infty]}\}_{i=1}^{\infty}$ that is a basis for $\mathcal{F}$. $\square$

Following lemma makes sure we can circumvent assumptions made in Korda & Mezić (2018b), which requires on the dictionary in the EDMD algorithm to be an orthonormal basis (o.n.b.) of $\mathcal{F}$.

**Lemma 3.** *Let $H, H'$ be the dataset used in Equation (12). For every set of $M \in \mathbb{N}$ linearly independent functions $\mathcal{F}_M = \{\phi_i\}_{i=1}^{M}$ from a dense subset of $\mathcal{F}$ and any function $f = c^{\mathsf{T}} \mathcal{F}_M$, there exists a $\tilde{c}$ and matrix $V$ such that*

$$\tilde{c}^{\mathsf{T}} \tilde{K} \tilde{\mathcal{F}}_M = c^{\mathsf{T}} K \mathcal{F}_M$$

*and*

$$\tilde{c}^{\mathsf{T}} \tilde{\mathcal{F}}_M = f = c^{\mathsf{T}} \mathcal{F}_M,$$

*where $\tilde{\mathcal{F}}_M = [\tilde{\psi}_1, \tilde{\psi}_2, \ldots, \tilde{\psi}_M]$ are functions from an orthornormal basis $\{\tilde{\psi}_i\}_{i=1}^{\infty}$ of $\mathcal{F}$, and $K, \tilde{K}$ are the Koopman approximations for the dictionaries $\mathcal{F}_M$ and $\tilde{\mathcal{F}}_M$ respectively.*

*Proof.* As $\mathcal{F}$ is a separable Hilbert space and a metric space, there exists a countable basis $\{\phi_i\}_{i=1}^{M} \cup \{\phi_i\}_{i=M+1}^{\infty}$, and by applying the Gram-Schmidt process to the basis, we have an o.n.b. $\{\tilde{\psi}_i\}_{i=1}^{\infty}$. Any $M$ step Gram-Schmidt process applied to a finite set of linearly independent vectors, can be written as a sequence of invertible matrices $V = \prod_{j=1}^{M+1} V_j$. Each matrix $V_j$ for $j < M + 1$ transforms the $j$th vector and the last matrix simply scales. Constructing such matrix $V$ applied to $\mathcal{F}_M$ yields $\tilde{\mathcal{F}}_M$. Setting $\tilde{c}^{\mathsf{T}} = c^{\mathsf{T}} V^{-1}$, which means

$$\tilde{c}^{\mathsf{T}} \tilde{\mathcal{F}}_M = c^{\mathsf{T}} V^{-1} \tilde{\mathcal{F}}_M = c^{\mathsf{T}} \mathcal{F}_M = f.$$

Furthermore, we have

$$\begin{aligned}
\tilde{c}^{\mathsf{T}} \tilde{K} \tilde{\mathcal{F}}_M &= c^{\mathsf{T}} V^{-1} [\tilde{\mathcal{F}}_M(H') \tilde{\mathcal{F}}_M(H)^+] V \mathcal{F}_M \\
&= c^{\mathsf{T}} V^{-1} [V \mathcal{F}_M(H') \mathcal{F}_M(H)^+ V^{-1}] V \mathcal{F}_M \\
&= c^{\mathsf{T}} [\mathcal{F}_M(H') \mathcal{F}_M(H)^+] \mathcal{F}_M = c^{\mathsf{T}} K \mathcal{F}_M.
\end{aligned}$$

$\square$

We are now ready to prove the Theorem 1 from the paper, namely the existence of networks for finite horizon predictions. We denote $\mathcal{F}^K$ as the space of vector valued functions functions $f = [f_1, f_2, \ldots, f_K]$, where $f_i \in \mathcal{F}$ and $\|f\| = \sum_{k=1}^{K} \|f_k\|_{L^2}$. In addition, we let $K_N$ be the Koopman approximation for $\mathcal{F}_M$ where $N$ data points have been used to solve the least square problem.

**Theorem 2.** *Let $f \in \mathcal{F}^K$, $H, H'$ be the dataset with $N$ data points used in Equation (12), and Assumption 2 and Assumption 3 hold. For any $\epsilon > 0$ and $T \in \mathbb{N}$, there exist an $M \in \mathbb{N}$ and hidden layers $\mathcal{F}_M$ with $M$ neurons and matrices $C$ such that*

$$\lim_{N \to \infty} \int_{\mathcal{H}} \|C K_N^t \mathcal{F}_M - f \circ F^t\|_2^2 d\mu_h < \epsilon, \tag{14}$$

*for all $t \in [1, 2, \ldots, T]$. In particular, there exist hidden layers and matrices $C$ such that*

$$\lim_{N \to \infty} \int_{\mathcal{H}} \|C K_N^t \mathcal{F}_M - F^t\|_2^2 d\mu_h < \epsilon. \tag{15}$$

*Proof.* W.l.o.g., we let $K = 1$. Due to Lemma 2, we know there exist hidden layers $\mathcal{F}_M$ and vectors $\boldsymbol{c}$ such that $\|f_m - f\|_{L^2}^2 < \epsilon_2$, where $\boldsymbol{c}^\mathsf{T} \mathcal{F}_M = f_m$ and

$$\epsilon_2 < \frac{\epsilon}{2 \cdot \max\{\|\mathcal{K}\|_{op}^{2T}, \|\mathcal{K}\|_{op}^2\}},$$

with $\|\cdot\|_{op}$ being the operator norm. This is possible due to Assumption 3 and Definition 2. We then have for any $t \in [1, 2, \ldots, T]$

$$\lim_{N \to \infty} \int_{\mathcal{H}} \|\boldsymbol{c}^\mathsf{T} K_N^t \mathcal{F}_M - \mathcal{K}^t f\|_2^2 d\mu_h$$

$$\leq \lim_{N \to \infty} \int_{\mathcal{H}} \|\boldsymbol{c}^\mathsf{T} K_N^t \mathcal{F}_M - \mathcal{K}^t f_m\|_2^2 d\mu_h + \|\mathcal{K}^t f_m - \mathcal{K}^t f\|_{L^2}$$

$$\leq \lim_{N \to \infty} \int_{\mathcal{H}} \|\boldsymbol{c}^\mathsf{T} K_N^t \mathcal{F}_M - \mathcal{K}^t f_m\|_2^2 d\mu_h + \|f_m - f\|_{L^2}^2 \max\{\|\mathcal{K}\|_{op}^{2T}, \|\mathcal{K}\|_{op}^2\}$$

$$< \frac{\epsilon}{2} + \frac{\epsilon}{2} = \epsilon,$$

where we use Theorem 5 in Korda & Mezić (2018b) to bound $\|\boldsymbol{c}^\mathsf{T} K_N^t \mathcal{F}_M - \mathcal{K}^t f_m\|_2^2 d\mu_h$ (in theory we might need a larger $M$, which we simply set and the bound of $f_m - f$ still holds). From the convergence above, Equation (14) follows by definition of the Koopman operator. For Equation (15), simply note that $f(\boldsymbol{h}) = \boldsymbol{h}$ is in $\mathcal{F}^{d_h}$ due to Definition 2, and the result holds. $\qquad\square$

### B.2 Controlled systems

Proving convergence for controlled systems gives different challenges. The results above cannot easily be shown for controlled systems. The reason being that the dictionary space one use is not a basis for the observables in the controlled setting, with the dynamical system extended by the left-shift operator, and the simplification made for EDMD in controlled systems. The results above may be extended, but the EDMD will not converge to the Koopman operator, but rather to $P_\infty^\mu \mathcal{K}_{\mathcal{F}_\infty}$, where $P_\infty^\mu$ is the $L^2(\mu)$ projection onto the closure of the dictionary space (Korda & Mezić, 2018a). However, results exists for the continuous controlled systems, with certain convergences for the generator. This is sadly not as strong as above, but an interesting path to connect RNNs/NeuralODEs to such theory (Nüske et al., 2023).

## C Interpretability using Koopman and sampling

Figure 7 shows two possible ways we can interpret our constructed RNN models, beyond what is usually possible for classical RNNs trained with gradient descent. On the left, we indicate which data pairs were chosen from the training set to construct neurons of the non-linear network $\mathcal{F}$. This can help to see if the "coverage" of the training set by neurons (resp. their associated data pairs) is reasonable, or if more neurons or data points are needed to cover highly non-linear regions.

On the right in Figure 7, we plot the locations of the eigenvalues of the Koopman matrix $K$ in our sampled RNN from Section 4.1. We can see that all eigenvalues are located on and inside the unit circle, indicating stable and oscillatory behavior.

## D ReLU-based RNNs for Dynamical Systems Modeling (DSM)

Most RNNs are parameterized discrete-time recursive maps of the given in Definition 1,

$$\boldsymbol{h}_t = F_{hx}(F_\mathcal{H}(\boldsymbol{h}_{t-1}), F_\mathcal{X}(\boldsymbol{x}_t)), \tag{16}$$

with latent states $\boldsymbol{h}_{t-1}$, optional external inputs $\boldsymbol{x}_t$. A piecewise linear RNN (PLRNN), introduced by Koppe et al. (2019), has the generic form

$$\boldsymbol{h}_t = W_h^{(1)} \boldsymbol{h}_{t-1} + W_h^{(2)} \sigma(\boldsymbol{h}_{t-1}) + \boldsymbol{b}_0 + W_x \boldsymbol{x}_t, \tag{17}$$

where $\sigma(\boldsymbol{h}_{t-1}) = \max(0, \boldsymbol{h}_{t-1})$ is the element-wise rectified linear unit (ReLU) function, $W_h^{(1)} \in \mathbb{R}^{d_h \times d_h}$ is a diagonal matrix of auto-regression weights, $W_h^{(2)} \in \mathbb{R}^{d_h \times d_h}$ is a matrix of connection

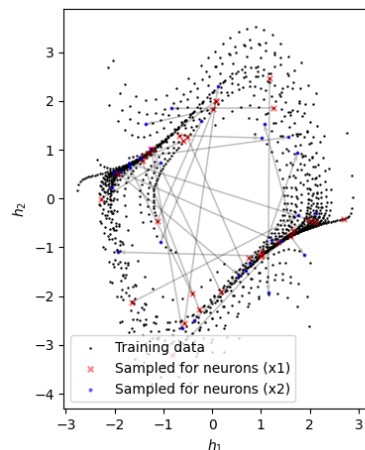 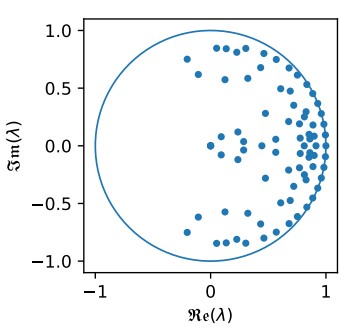

Figure 7: Left: Interpretability of the hidden state network weights over the data domain in the Van der Pol example. The arrows (with initial $x_1$ and final points $x_2$) depict the pairs that were chosen during sampling. This means that they directly visualize where neurons are placed on the domain. Right: Eigenvalues of the Koopman matrix approximated with 80 neurons for the Van der Pol example.

weights, the vector $\boldsymbol{b}_0 \in \mathbb{R}^{d_h}$ represents the bias, and the external input is weighted by $W_x \in \mathbb{R}^{d_h \times d_x}$. Afterwards, Brenner et al. (2022) extended this basic structure by incorporating a linear spline basis expansion, referred to as the dendritic PLRNN (dendPLRNN)

$$\boldsymbol{h}_t = W_h^{(1)} \boldsymbol{h}_{t-1} + W_h^{(2)} \sum_{j=1}^{J} \alpha_j \, \sigma(\boldsymbol{h}_{t-1} - \boldsymbol{b}_j) + \boldsymbol{b}_0 + W_x \boldsymbol{x}_t, \tag{18}$$

where $\{\alpha_j, \boldsymbol{b}_j\}_{j=1}^J$ represents slope-threshold pairs, with $J$ denoting the number of bases. This expansion was introduced to increase the expressivity of each unit's nonlinearity, thereby facilitating DSM in reduced dimensions. Moreover, Hess et al. (2023) proposed the following "1-hidden-layer" ReLU-based RNN, which they referred to as the shallow PLRNN (shPLRNN)

$$\boldsymbol{h}_t = W_h^{(1)} \boldsymbol{h}_{t-1} + W_h^{(2)} \sigma(W_h^{(3)} \boldsymbol{h}_{t-1} + \boldsymbol{b}_1) + \boldsymbol{b}_0 + W_x \boldsymbol{x}_t, \tag{19}$$

where $W_h^{(1)} \in \mathbb{R}^{d_h \times d_h}$ is a diagonal matrix, $W_h^{(2)} \in \mathbb{R}^{d_h \times M}$ and $W_h^{(3)} \in \mathbb{R}^{M \times d_h}$ are rectangular connectivity matrices, and $\boldsymbol{b}_1 \in \mathbb{R}^M$, $\boldsymbol{b}_0 \in \mathbb{R}^{d_h}$ denote thresholds. The combination of Generalized Teacher Forcing (GTF) and shPLRNN results in a powerful DSM algorithm on challenging real-world data; for more information see Hess et al. (2023). When $M > d_h$, it is possible to rewrite any shPLRNN as a dendPLRNN by expanding the activation of each unit into a weighted sum of ReLU nonlinearities (Hess et al., 2023).

A clipping mechanism can be added to the shPLRNN to prevent states from diverging to infinity as a result of the unbounded ReLU nonlinearity

$$\boldsymbol{h}_t = W_h^{(1)} \boldsymbol{h}_{t-1} + W_h^{(2)} \big[ \sigma(W_h^{(3)} \boldsymbol{h}_{t-1} + \boldsymbol{b}_1) - \sigma(W_h^{(3)} \boldsymbol{h}_{t-1}) \big] + \boldsymbol{b}_0 + W_x \boldsymbol{x}_t. \tag{20}$$

This guarantees bounded orbits under certain conditions on the matrix $W_h^{(1)}$ (Hess et al., 2023).

In our experiments with RNNs, the clipped shPLRNN is trained by GTF for DSM on the benchmark systems (see below Appendix F).

## E    RESERVOIR MODELS: ECHO STATE NETWORKS

As our newly proposed method bears similarities to a reservoir computing architecture, we have trained reservoir models as part of our computational experiments. We used the simplest recurrent

reservoir architecture, which is an echo state network (ESN) introduced by Jaeger & Haas (2004). Here we will briefly introduce the main ideas behind ESNs, and we refer the interested reader to the review by Lukoševičius & Jaeger (2009) for more details.

An ESN consists of a reservoir and a readout. The reservoir contains neurons which are randomly connected to inputs and are these are not trained. Denoting the inputs as $\boldsymbol{h}_t \in \mathbb{R}^{N_h}$ and output as $\boldsymbol{y}_t \in \mathbb{R}^{N_y}$, and the internal reservoir states as $\boldsymbol{k}_t \in \mathbb{R}^{N_k}$, the reservoir provides an update rule for the internal units as

$$\boldsymbol{k}_{t+1} = f\left(W^{in}\boldsymbol{h}_{t+1} + W\boldsymbol{k}_t + W^{back}\boldsymbol{y}_t\right), \qquad (21)$$

for an activation function $f$ and weight matrices $W^{in} \in \mathbb{R}^{N_k \times N_h}$, $W \in \mathbb{R}^{N_k \times N_k}$ and $W^{back} \in \mathbb{R}^{N_k \times N_y}$. After the reservoir comes the readout, which maps the inputs, reservoir states and outputs to a new output state

$$\boldsymbol{y}_{t+1} = f_{out}\left(W^{out}(\boldsymbol{h}_{t+1}, \boldsymbol{k}_{t+1}, \boldsymbol{y}_t)\right), \qquad (22)$$

where $f^{out}$ is the output activation, $W^{out} \in \mathbb{R}^{N_y \times N_y}$ are output weights and $(\boldsymbol{h}_{t+1}, \boldsymbol{k}_{t+1}, \boldsymbol{y}_t)$ denotes the concatenation of $\boldsymbol{h}_{t+1}$, $\boldsymbol{k}_{t+1}$ and $\boldsymbol{y}_t$. In the readout the model learns the connections from the reservoir to the readout, for example via (regularized) regression. A so-called feedback connection allows for the readout values to be fed back into the reservoir, as shown in Equation (21), establishing a recurrent relation.

## F  BENCHMARK SYSTEMS AND REAL-WORLD DATA

**Van der Pol**   In 1927, Balthasar Van der Pol introduced a non-conservative oscillatory system with a nonlinear damping term to describe oscillations in a vacuum tube electrical circuit. The system is described as a two dimensional ODE

$$\begin{cases} \dot{h_1} &= h_2 \\ \dot{h_2} &= \mu(1 - h_1^2) - h_1 + h_2, \end{cases}$$

where $\mu$ is a scalar parameter indicating the nonlinearity and the strength of the damping. In our experiment we set $\mu = 1$, that is, in the limit cycle regime. For the experiments in Section 4.1 and Section 4.2, training data are generated by solving an initial value problem for $t \in [0, 20]$ with $\Delta t = 0.1$ for 50 initial conditions, where each initial condition is random vector $\boldsymbol{h}_0 \sim \text{Uniform}([-3, 3]^2)$. We used an explicit Runge-Kutta method of order 8 to solve the initial value problem. Validation and test data are generated similarly but for $t \in [0, 50]$.

**Lorenz-63**   Devised by Edward Lorenz in 1963 (Lorenz, 1963) to model atmospheric convection, the Lorenz-63 system is defined as

$$\begin{cases} \dot{h_1} &= \sigma(h_2 - h_1) \\ \dot{h_2} &= h_1(\rho - h_3) - h_2 \\ \dot{h_3} &= h_1 h_2 - \beta h_3, \end{cases}$$

where $\sigma, \rho, \beta$, are parameters that control the dynamics of the system. In our experiment, we set $\sigma = 10$, $\beta = \frac{8}{3}$, and $\rho = 28$, which means we are in the chaotic regime. For the experiments in Section 4.3, training data are generated by solving an initial value problem for $t \in [0, 5]$ with $\Delta t = 0.01$ for 50 initial conditions, where each initial condition is random vector $\boldsymbol{h}_0 \sim \text{Uniform}([-20, 20] \times [-20, 20] \times [0, 50])$. The solver used explicit Runge-Kutta of order 8, likewise as the previous dataset. Validation and test data are generated similarly but for $t \in [0, 50]$. We normalize datasets to scale the values to the range $[-3, 3]$ to improve the training.

**Rössler**   Otto Rössler introduced the Rössler system in 1976 (Rössler, 1976) as a model that generates chaotic dynamics

$$\begin{cases} \dot{h_1} &= -h_2 - h_3 \\ \dot{h_2} &= h_1 + \alpha h_2 \\ \dot{h_3} &= \beta + h_3(h_1 - \kappa), \end{cases}$$

where $\alpha$, $\beta$, $\kappa$, are parameters controlling the dynamics of the system. Here, we set $\alpha = 0.15$, $\beta = 0.2$, and $\kappa = 10$, which puts the system in the chaotic regime. The setup for data generation is similar to the Lorenz example. For the experiments in Table 1, training data are generated by solving an initial value problem for $t \in [0, 10]$ with $\Delta t = 0.01$ for 50 initial conditions, where each initial condition is random vector $\boldsymbol{h}_0 \sim \text{Uniform}([-20, 20] \times [-20, 20] \times [0, 40])$. Again, an explicit Runge-Kutta method of order 8 was used. Validation and test data are generated similarly but for $t \in [0, 200]$. We normalize datasets to scale the values to the range $[-3, 3]$ to improve the training.

**Forced Van der Pol Oscillator**   As an example for a controlled system, we use the Van Der Pol oscillator with external input forcing $x$, and

$$\begin{cases} \dot{h_1} & = h_2 \\ \dot{h_2} & = \mu(1 - h_1^2)h_2 - h_1 + x \end{cases}$$

The data is obtained for $t \in [0, 50\Delta t]$ with $\Delta t = 0.05$, with 150 initial conditions, where $\boldsymbol{h}_0 \sim \text{Uniform}([-3, 3]^2)$ and $x_0 \sim \text{Uniform}([-3, 3])$. The solver used here was using an explicit Runge-Kutta method of order 5(4). It is important to highlight that the control input data $x_0$ does not come from any controller with a particular target state, i.e. random control is applied to the trajectories in the training dataset.

**Weather data**   In this experiment, we follow TensorFlow (2024) and use the Jena Climate dataset (for Biogeochemistry, 2024). The original data contains inconsistent date and time values, leading to gaps and overlaps between measurements. We extracted the longest consecutive time period and thus worked with the data between July 1st, 2010, and May 16th, 2013. We additionally downsampled the time series from the original 10-minute to 1-hour measurements. Then, the first 70% of records were used as the train set, the next 20% as the validation, and the remaining 10% as the test set. Identically to TensorFlow (2024), we pre-processed the features and added `sin` and `cos` time-embeddings of hour, day, and month. We plot the dataset, indicating the train-validation-test split with colors in Figure 8.

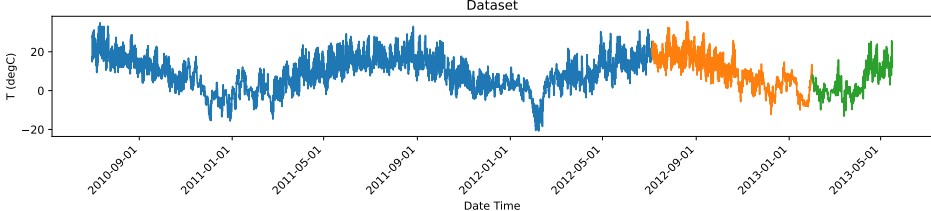

Figure 8: The weather dataset and its splits: train (blue), validation (orange), and test (green).

## G   EVALUATION MEASURES

### G.1   GEOMETRICAL MEASURE

The Kullback–Leibler divergence of two probabiliti densities $p(\boldsymbol{x})$ and $q(\boldsymbol{x})$ is defined as

$$D_{KL}(p(\boldsymbol{x}) \| q(\boldsymbol{x})) = \int_{\boldsymbol{x} \in \mathbb{R}} p(\boldsymbol{x}) \log \frac{p(\boldsymbol{x})}{q(\boldsymbol{x})} d\boldsymbol{x}. \tag{23}$$

In order to be able to accurately evaluate also high-dimensional systems, we follow the approach used in Hess et al. (2023) and place Gaussian Mixture Models (GMM) on the along the true trajectory $\boldsymbol{x}$ and predicted $\hat{\boldsymbol{x}}$ trajectories, obtaining $\hat{p}(\boldsymbol{x}) = \frac{1}{T} \sum_{t=1}^{T} \mathcal{N}(\boldsymbol{x}, \boldsymbol{x}_t, \Sigma)$ and $\hat{q}(\boldsymbol{x}) = \frac{1}{T} \sum_{t=1}^{T} \mathcal{N}(\boldsymbol{x}, \hat{\boldsymbol{x}}_t, \Sigma)$ for $T$ snapshots. Using the estimated densities, we consider a Monte Carlo approximation of Equation (23) by drawing $n$ random samples from the GMMs and obtain the density measure

$$D_{KL}(\hat{p}(\boldsymbol{x}) \| \hat{q}(\boldsymbol{x})) \approx \frac{1}{n} \sum_{i=1}^{n} \log \frac{\hat{p}(\boldsymbol{x})}{\hat{q}(\boldsymbol{x})}. \tag{24}$$

We call this metric empirical KL divergence (EKL) in the manuscript. To make our results comparable with Hess et al. (2023), we use $\sigma^2 = 1.0$ and $n = 1000$.

# H  MODEL DETAILS AND COMPARISON

## H.1  SAMPLED RNN

The implementation of our gradient-free training module for sampled RNNs was done in Python since the key tools for the algorithm already exist as Python libraries. The algorithm requires the ability to sample weights and biases, thus we used the Python library `swimnetworks` by Bolager et al. (2023). Furthermore, we were able to alleviate the approximation of the Koopman operator, in the uncontrolled as well as controlled setting using some functionalities from the Python library `datafold` by Lehmberg et al. (2020).

Sampled RNNs have only a few hyperparameters: the number of nodes in the hidden layer, the activation function of the hidden layer, and a cutoff for small singular values in the least-squares solver. We often refer to the singular value cutoff hyperparameter as the regularization rate. In the case of a sampled RNN with time delay, additional hyperparameters are the number of time delays and the number of PCA components, if used.

Sampled RNNs do not only have a low fit time but also a short hyperparameter tuning since there are only a few degrees of freedom. Thus, our newly proposed method offers efficiency beyond the short training time.

We empirically investigated the scaling of sampled RNNs as a function of the number of neurons. Similarly, as for the other results, we perform five runs and report the average as well as the minimum and maximum; the results can be seen in Figure 9. These are results for a dataset of fixed size, where we would expect that the only change in computation is the least-squares solver and additional samples of weights and biases.

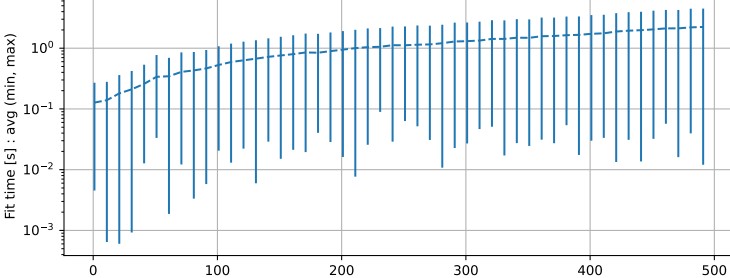

Figure 9: Fit time over number of neurons for the Van der Pol experiment from Section 4.1.

In addition, we investigated the effect of increasing the number of neurons on training and validation performance. Results from five runs are shown in Figure 10. We notice that the errors decrease up to a certain number of neurons and stagnate afterwards.

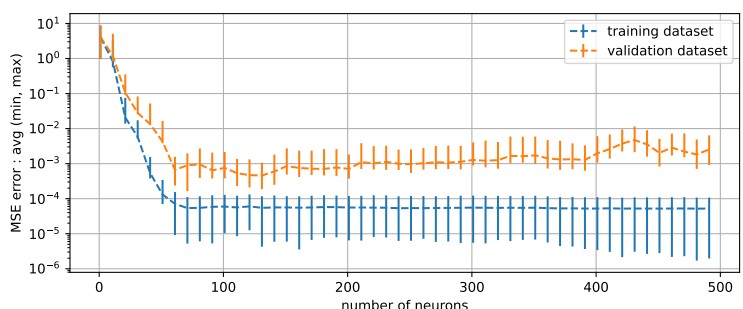

Figure 10: Training and validation MSE error over number of neurons for the Van der Pol experiment from Section 4.1.

The Van der Pol experiment from Section 4.1 was used as one of the first prototypes of our method. Our initial experiments used a smaller dataset than the one described in Appendix F, namely with very short trajectories consisting of only three time steps. The sampled RNN was able to learn the full dynamics from this data. However, it was not possible to train the gradient-based shPLRNN with such a dataset; thus, we used longer trajectories to be able to compare our method with others. The final choice of hyperparameters for the hyperparameters is given in Table 2.

Training the 1D Van der Pol from Section 4.2 was done similarly, the number of hidden nodes and activation function remained unchanged. The hyperparameters are listed in Table 2. A time delay of six was chosen, however it was also possible to use only two time delays and get an excellent performance, but we opted against this choice because this was based on our knowledge that the true system is two dimensional. Thus, we opted for a higher number of time delays which then get reduced with the additional PCA transformation to mimic a real-world scenario where the true dimension of the problem is unknown.

The chaotic systems from Section 4.3 required more hidden nodes in order to obtain good prediction, as compared to the Van der Pol but nonetheless the process of hyperparameter tuning was simple. The final choice of hyperparameters for the Lorenz and Rössler problems is given in Table 2. We show a visual comparison of the predicted trajectories from the sampled RNNs, ESN and shPLRNNs for the Lorenz system in Figure 12.

We also considered how the number of hidden units effects the model's performance. For the number of nodes considered powers of two, specifically $2^2$ until $2^9$ for both chaotic models. Each run was repeated five times to provide errorbars from min to max in addition to the average value, show in Figure 11. We notice the typical trend of a decreasing error, with possible overfitting in the case of Rossler.

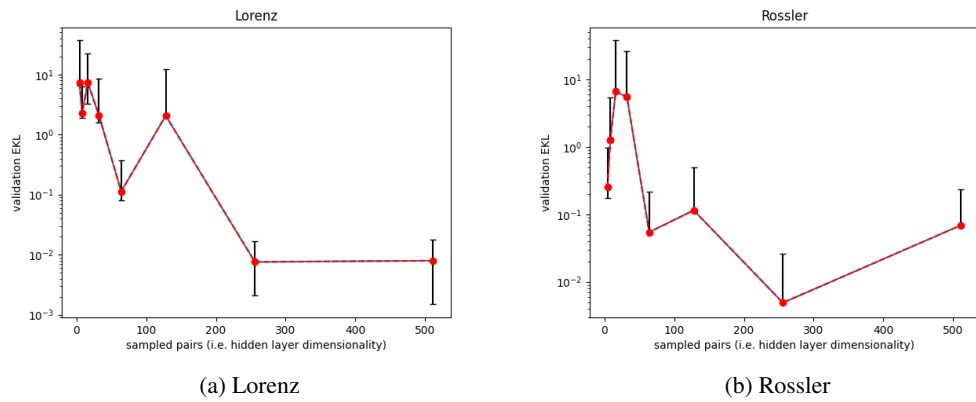

(a) Lorenz

(b) Rossler

Figure 11: Ablation of hidden units for chaotic systems.

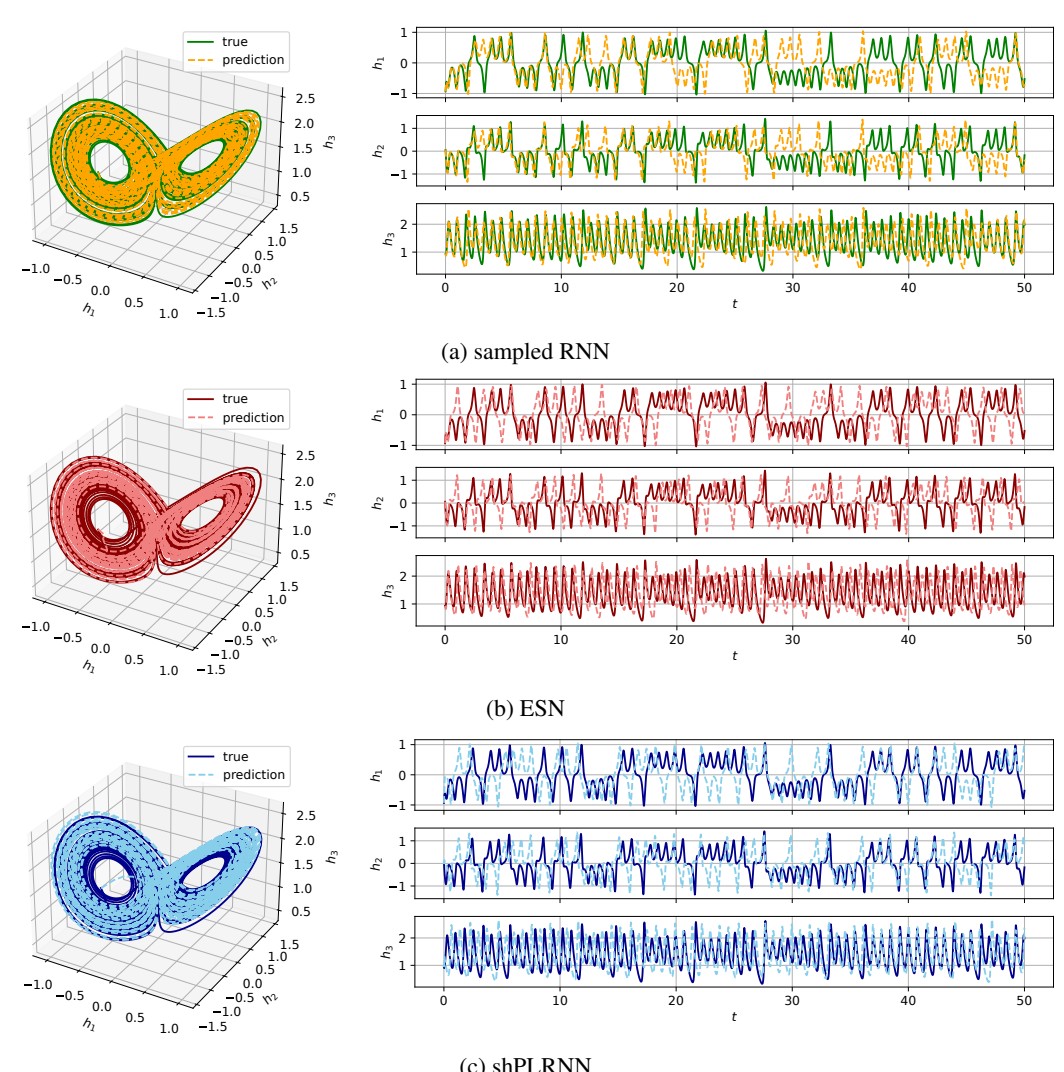

(a) sampled RNN

(b) ESN

(c) shPLRNN

Figure 12: Visual comparison of different models on the same test trajectory.

With the forced Van der Pol example, presented in Section 4.4, we use the sampled RNN as a surrogate model to a controller, essentially making this a model-based control example. In order to be able to accurately control the state, the surrogate needs to capture the dynamics of the system with sufficient accuracy. The hyperparameters are listed in Table 2. We opted for an LQR controller as our optimal controller, as a simple choice for linear systems. Although the Van der Pol oscillator is not a linear system, the surrogate sampled RNN captures the dynamics via a linear map with the Koopman operator, allowing minimizing the cost. For our LQR, $R = 1$, and $Q = \mathrm{diag}(10)$. The dataset we use contains randomly initialized trajectories evolved in time, as well as randomly chosen control inputs. We observed that a dataset with many shorter trajectories is more useful than a dataset with a few longer trajectories since for this problem diverse initial conditions are more informative of the vector field of the system, as opposed to long trajectories which become periodic. The necessary effort for tuning the sampled RNN hyperparameters and LQR controller parameters was low.

Although not reported in the manuscript as part of the experiments, in Appendix H.1.1 we discuss the forced Van der Pol model using a nonlinearity for the inputs.

The training of a sampling RNN on weather data was performed using time-delay embeddings to account for a possibly partial observation of the state. The objective is to train a model which can predict the temperature. To find the hyperparameters a grid search was performed, and this is

documented in Table 7. The final choice of hyperparameters is given in Table 2. After training, predictions are done for fixed chunks of time, which are concatenated together afterwards. The number of time-delays dictates how many ground-truth datapoints are necessary to predict the next state, which gets concatenated with the previous predictions. We see this as a reasonable approach for weather prediction, as typically one would use the available information for a fairly long period of time, and predict for a fixed, likely shorter time horizon.

### H.1.1 LQR with nonlinear lifting

In this section we describe shortly how we perform model predictive control described in Algorithm 4 with nonlinear lifting $\mathcal{G}_{\hat{M}}$, i.e. applying a sampling hidden layer on the input $x$. The experiment in the main paper was mainly done by setting $\mathcal{G}_{\hat{M}}$ to the identity, but adding nonlinearity with $\hat{M} > d_x$ is possible and may beneficial for future testing, even though for the particular system in the main paper, it yielded equivalent results as with $\mathcal{G}_{\hat{M}} \equiv Id$.

Following Algorithm 4, we lift $\boldsymbol{x}_t$ to $\mathbb{R}^{\hat{M}}$ through the hidden layer $\mathcal{G}_{\hat{M}}$. Then we solve for $B$ exactly as described in Algorithm 2 and fit the LQR. The LQR computes the control input, but now in the $\mathbb{R}^{\hat{M}}$ space. Before we can pass it to $F$, we approximate a linear map $P$, such that $P\mathcal{G}_{\hat{M}}(\boldsymbol{x}_t) \approx \boldsymbol{x}_t$. Once projected down, we can pass the projected value $\hat{\boldsymbol{x}}_t$ to $F$ and continue as usual.

### H.1.2 Hyperparameters

Table 2: Hyperparameters of sampled RNN models

|  | Van der Pol | 1D Van der Pol | Lorenz | Rössler | force Van der Pol | Weather |
|---|---|---|---|---|---|---|
| Hidden layer width | 80 | 80 | 200 | 300 | 128 | 256 |
| Activation function | $tanh$ | $tanh$ | $tanh$ | $tanh$ | $tanh$ | $tanh$ |
| Regularization rate | 1e-8 | 0 | 1e-7 | 1e-4 | 1e-10 | 1e-6 |
| Time delays | - | 6 | - | - | - | 168 |
| PCA components | - | 2 | - | - | - | - |

### H.1.3 Hardware

The machine used for training the sampled RNNs was 13th Gen Intel(R) Core(TM) i5-1335U @ 4.6 GHz (16GB RAM, 12 cores), no GPU hardware was used.

## H.2 ESN

For the implementation of ESNs we used the Python library `reservoirpy` by Trouvain et al. (2020). All models were trained using a single reservoir and a ridge regression readout. We consider only reservoir models without any warm-up phase in order to keep them comparable to our sampling RNN which does not have a warm-up.

For the chaotic systems, we were able to find guidelines in the literature for a suitable choice of hyperparameters specifically tailored to the Lorenz system (see (Viehweg et al., 2023)). With minor modifications and following the proposed guidelines, we found hyperparameters also for the Rössler system. The choice of hyperparameters is specified in Table 3, any hyperparameters not mentioned are set to the default value of `reservoirpy`.

For the Van der Pol problem many hyperparameter combinations were tried out until we were able to find a model with good performance and sufficient robustness to a change in the random seed. The hyperparameter combinations we considered are shown in Table 4. Due to the high expenses of a grid-search approach, a random search was employed and 1000 hyperparameter combinations were considered. The hyperparameter choice with the best performance on the validation dataset was selected and then evaluated on the test dataset. The final choice of hyperparameters is documented in Table 3, and any unmentioned hyperparameters are assumed to be set to the default value from the `reservoirpy` library.

### H.2.1 Hyperparameters

Table 3: Hyperparameters of reservoir models

|  | Van der Pol 4.1 | Lorenz 4.3 | Rössler 4.3 |
|---|---|---|---|
| Width/units | 500 | 300 | 500 |
| Leak rate | 0.9 | 0.3 | 0.3 |
| Spectral radius | 0.5 | 1.25 | 0.5 |
| Input scaling | 0.05 | 0.1 | 0.1 |
| Connectivity | 0.8 | 0.1 | 0.1 |
| Inter connectivity | 0.2 | 0.2 | 0.2 |
| Ridge regularization coeff. | 1e-10 | 1e-4 | 1e-8 |
| Warmup steps | 0 | 0 | 0 |

Table 4: Hyperparameters used in random search on a grid for a Van der Pol reservoir model

|  | Van der Pol 4.1 |
|---|---|
| Width/units | 100, 200, 500 |
| Leak rate | 0.1, 0.3, 0.5, 0.7, 0.9 |
| Spectral radius | 0.25, 0.5, 0.75, 1, 1.25, 1.5, 2, 3, 5 |
| Input scaling | 0.05, 0.1, 0.5, 1, 1.5, 2 |
| Connectivity | 0.2, 0.4, 0.6, 0.8, 1 |
| Inter connectivity | 0.2, 0.4, 0.6, 0.8, 1 |
| Ridge regularization coeff. | 1e-4, 1e-6, 1e-8, 1e-10 |
| Warmup steps | 0 |

### H.2.2 Hardware

The machine used for fitting the ESN models was 13th Gen Intel(R) Core(TM) i5-1335U @ 4.6 GHz (16GB RAM, 12 cores).

### H.3 shPLRNN

For an explanation of shPLRNN, see Appendix D.

### H.3.1 Hyperparameters

We used the clipped shPLRNN trained by GTF. For the Van der Pol, Lorenz and the weather datasets we considered a fixed GTF parameter $\alpha$, while for the Rössler we considered an adaptive $\alpha$ (starting from an upper bound) as proposed by (Hess et al., 2023). The code repository by (Hess et al., 2023) was used to perform the computations. The hyperparameters selected for all datasets are detailed in Table 5. Any hyperparameters not specified are set to their default values in the corresponding repository of (Hess et al., 2023). For Rössler dataset, finding optimal hyperparameters was more challenging, and for training, we also utilized regularizations for the latent and observation models.

Table 5: Hyperparameters of shPLRNN trained by GTF

|  | Van der Pol 4.1 | Lorenz 4.3 | Rössler4.3 | Weather |
|---|---|---|---|---|
| Hidden dimension | 35 | 100 | 50 | 200 |
| Batch Size | 32 | 30 | 50 | 32 |
| Sequence length | 37 | 100 | 150 | 40 |
| Teacher forcing interval | 25 | 13 | 25 | 20 |
| Epochs | 1300 | 2000 | 2000 | 2500 |
| GTF parameter $\alpha$ | 0.98 | 0.3 | 0.9 (Upper bound) | 0.75 |
| Latent model regularization rate | - | - | 1e-6 | - |
| Observation model regularization rate | - | - | 1e-4 | - |

### H.3.2 HARDWARE

The hardware we used to iteratively train the clipped shPLRNN models includes an 11th Gen Intel(R) Core(TM) i7-11800H CPU @ 2.30GHz and 64.0 GB of RAM (63.7 GB usable).

## H.4 LSTM

We use Tensorflow (Abadi et al., 2015) to train a baseline LSTM for the weather data problem of predicting the temperature. An LSTM is trained similarly to the process by TensorFlow (2024).

After training the weather models, we compute the predictions with the specified horizon and concatenate them into a single time series. More specifically, if the horizon is set to one day with a time delay of one week, the prediction on a dataset split goes as follows. First, we use ground-truth values of days one to seven to predict the value for day eight. Then, we use the ground-truth values of days two to eight to predict the value for day nine. This process is repeated until we reach the end of the split. Then, the resulting predictions are concatenated and compared to the ground-truth measurements. This prediction process alligns with the one used for sampled RNNs to ensure a consistent comparison.

### H.4.1 HYPERPARAMETERS

Table 6: Hyperparameters used for LSTM for Section 4.5.

|  | LSTM |
| --- | --- |
| Width/units | 64 |
| Learning rate | 5e-5 |
| Max epochs | 30 |
| Patience | 5 |

See Table 7 for details on the hyperparameter grid search for Section 4.5 performed for the sampled RNN and LSTM models.

Table 7: Hyperparameters used in the grid search for training sampled RNN and LSTM for the temperature prediction in Section 4.5.

|  | sampled RNN | LSTM |
| --- | --- | --- |
| Width/units | 32, 64, 128 | 32, 64, 128, 256, 512 |
| Regularization rate | 1e-10, 1e-8, 1e-6, 1e-4, 1e-2 | — |
| Learning rate | — | 1e-5, 5e-5, 1e-4, 5e-4, 1e-3, 5e-3 |

### H.4.2 HARDWARE

For the experiments with the weather data in Section 4.5, we used a machine with AMD EPYC 7402 @ 2.80GHz (256GB RAM, 24 cores) and RTX 3080 Turbo (10GB VRAM, CUDA 12.0).

## I ADDITIONAL EXPERIMENT WITH REAL-WORLD DATA

We provide a additional experiments to showcase our model's performance on real world data. From our experience it was not necessary to perform exhaustive hyperparamter tuning in order to obtain decent performance from a sampled RNN model. The same hardware was used as for the weather data experiment mentioned in Appendix H.4.2.

**Electricity consumption** We use the individual power consumption dataset by Hebrail & Berard (2006) and predict the voltage feature. This dataset contains measurements made in a one-minute interval, and we consider a period of four weeks as our dataset. Two weeks are used as training data, one week as validation and one as test data, as shown in Figure 13. We added `sin` and `cos`

time-embeddings of hour and day. We use 240 time-delay embeddings (amounting to 4 hours) and always predict a horizon of 120 steps (2 hours). The sampled RNN has 128 hidden nodes and a $tanh$ activation. The results are plotted in Figure 14. Training time was 6.5 seconds. The MSE error on training, validation and test data is 1.565, 1.427 and 1.428 V respectively.

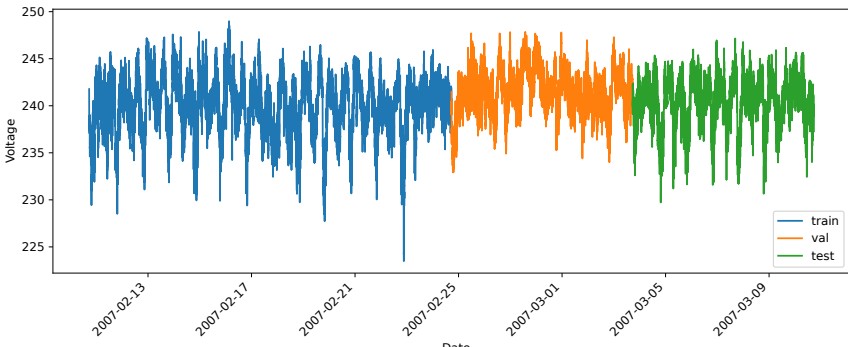

Figure 13: Individual household electrical power consumption data split into training, validation and testing portions.

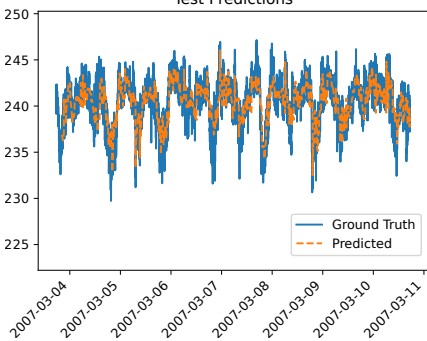

Figure 14: Test set predictions of individual household electricity consumption.

**Bike traffic** We use a dataset containing bike traffic counts in Copenhagen bik. This dataset contains only the bike count and the date and time information. We use the bike count as a target variable. We added `sin` and `cos` time-embeddings of hour, day, and month. The largest portion of data without missing values is used as the dataset, and split into training, validation and test as shown in Figure 15. We use a sampled RNN with 256 hidden nodes and $tanh$ activation. A time delay of 14 hours is used, and the prediction horizon is 7 hours. Training time was 0.65 seconds. The MSE error on the training, validation and test set is 90, 160, 140, respectively.

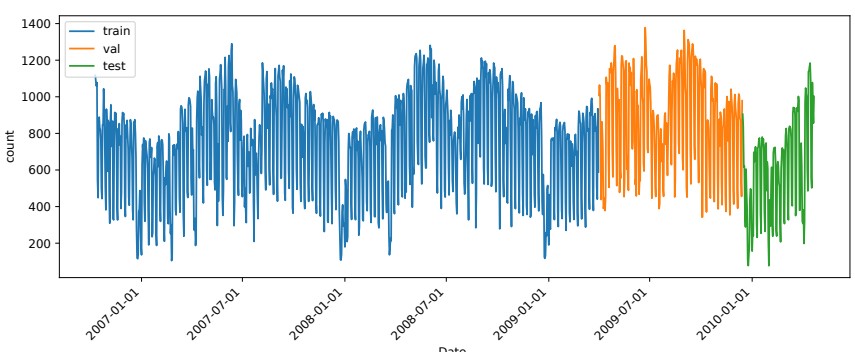

Figure 15: Bike count data split into training, validation and testing portions.

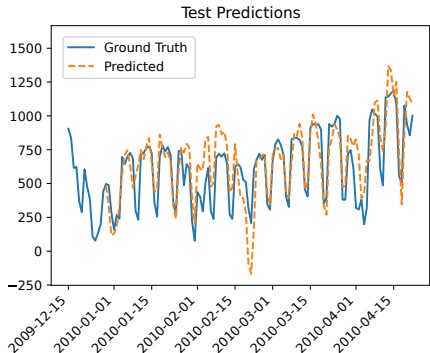

Figure 16: Test set predictions of bike counts.

## J    COMPARISON WITHOUT SWIM AND WITHOUT KOOPMAN

In this section, we report on some results performed to separate each contribution of the parts we use in this paper, namely the SWIM sampling strategy and the Koopman operator. We ran two settings (excluding our method already reported). One is with the addition of the Koopman operator and EDMD, but where we sample the weights of the hidden layer with a standard Gaussian distribution and the biases with a uniform distribution. This sampling strategy is commonly used for random neural networks, which is why we chose these distributions in particular. In the second setting, we exclude the EDMD step and simply project down the original state space after applying the hidden layer. The results can be found in Table 8. We observe that the model that incorporates both Koopman and SWIM performs best over all the experiments we tested, and with typically smaller error bars.

Table 8: Test setting (A): without SWIM and with Koopman, (B): with SWIM and without Koopman, (C) with SWIM and with Koopman (our). Results for Van der Pol are reported with MSE and the chaotic systems Lorenz and Rössler are reported with EKL, and all experiments report mean, minimum, and maximum respectively.

| Example | (A) | (B) | (C) |
|---|---|---|---|
| Van der Pol | 2.90e-3 (6e-4,8.1e-3) | 3.12e-2 (3.05e-2, 3.16e-2) | 9.55e-4 (7.08e-4, 1.28e-3) |
| Lorenz | 7.79e-3 (5.79e-3, 9.87e-3) | 8.41e-3 (6.45e-3, 1.02e-2) | 4.36e-3 (3.66e-3, 5.36e-3) |
| Rössler | 4.47e-1 (1.7e-4, 2.21e-0) | 7.51e-3 (1.00e-4,3.63e-2) | 1.57e-4 (5.86e-5, 3.82e-4) |

Finally, to see how our method compares to one without Koopman when one extends the size of the time step to predict the Van der Pol system, we ran an experiment for time steps $\Delta t =$

$[0.1, 0.2, 0.3, 0.4, 0.5]$, and the results can be found in Figure 17. Here, we observe that the method, including Koopman, consistently performs better and has a much more stable error bar as we increase $\Delta t$.

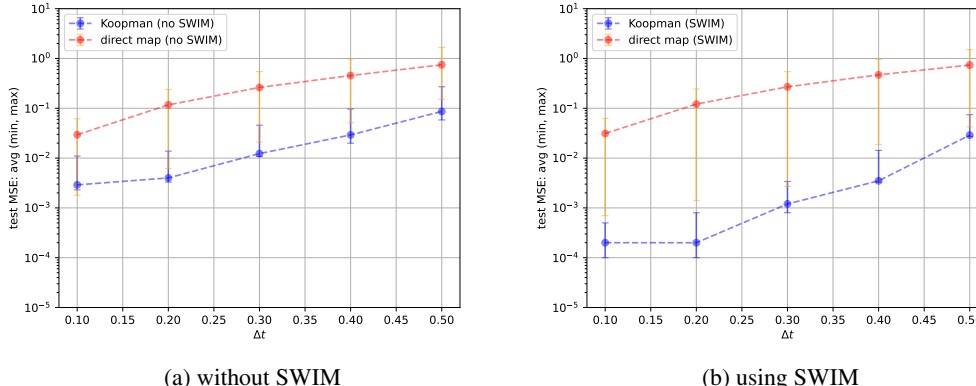

(a) without SWIM                    (b) using SWIM

Figure 17: Running our model and a model without Koopman while increasing the size of the time step the models predict, $\Delta t = [0.1, 0.2, 0.3, 0.4, 0.5]$, results reported on test set with MSE and min/max error bars.

