# OpenReview forum: "Gradient-free training of recurrent neural networks"
_ICLR.cc/2025/Conference — Submitted to ICLR 2025_

### Official Review · Reviewer_Bq2A · 2024-11-03

**Soundness:** 4
**Presentation:** 3
**Contribution:** 3
**Rating:** 8
**Confidence:** 4

**Summary:**

The MS explores an alternative method for fitting recurrent neural networks, based on Koopman operator theory.  A single-layer neural network with *random weights* is used to map the state (and possibly a control input) to a higher-dimensional space, in which the dynamics are assumed to be linear.  Since the state is assumed fully observed, the low-D state at consecutive time steps can be mapped to the high-D state, and the state matrix fit with the normal equations (and similarly for the input matrix when there is a control input).  The map back to the low-dimensional state and the output matrix can be found likewise.  The intuition from Koopman operator theory is that there exists a set, possibly infinite-dimensional, of measurement functions that evolve linearly in time.  The authors shows that this method yields computationally cheap and highly accurate models of simple nonlinear systems.

**Strengths:**

The proposed approach fits simple nonlinear systems with high accuracy and very little computational time compared with using BPTT in RNNs.  This makes the approach a very appealing alternative for modeling and controlling such systems.  To my knowledge, the approach is novel (but see below), and open up a new perpective on random networks.

**Weaknesses:**

From the point of view of implementation, the manuscript's random RNNs are a fairly minor variation on echo-state networks.  Indeed, there is a large literature on ESNs and reservoir computing, and I am surprised that this variation has not previously been explored.  Can the authors confirm this?

The introduction of Koopman operator theory does provide a nice intuition about why a linear dynamical system should exist in a higher-dimensional space.  But it seems that the trade-off it enables---linearity for higher dimension---limits the scope of application of this approach: As the authors note, the matrix inversion operation (in the normal equations) is expensive, and it is cubic precisely the dimension of the (large) measurement space.

My understanding of the theory is that, in general, not only is this space infinite-dimensional, but also there is no way to bound the number of required dimensions.  That is, M could be arbitrarily large.  Perhaps this is addressed in the proof in Appendix B, which I did not read closely.  Can the authors provide more insight here?

Finally, I found the paper somewhat hard to read.  This could be my fault, but there seem to be notational issues.  For example, near l. 139 the authors write h_t = F(h_{t-1}), and then later in the same paragraph, h_t' = F(h_t).  Is the idea that, in the first instance, h_t is the model state; where in the second instance, h_t is the *true* state (and therefore F(h_t) need not be equal to h_{t+1})?

**Questions:**

The authors compare against ESNs on some problems and LSTM on others.  Is there any reason for these choices?

Can the authors provide more intuition about the weight initialization, i.e., Eq. 4 and the equation for the distribution p_H?

---

> ### Author Response · Authors · 2024-11-23
>
> We would like to thank the reviewer for the review, and we believe certain points in the review have greatly strengthened the current manuscript. Below, we make a few comments on the weaknesses pointed out, as well as answer the reviewer's questions.
>
> [W1] Indeed, the connection between Koopman operator approximation methods and reservoir computing (RC) / echo-state networks (ESNs) has not been explored extensively. We are only aware of the two papers we cite in the section on related work (Bollt (2021); Gulina & Mauroy (2020)). Bollt (2021) only briefly mentions the connection of RC to DMD (i.e. without nonlinear observables), while Gulina & Mauroy (2020) use the hidden state of the reservoir as a dictionary to approximate the Koopman operator - but do not use the Koopman operator for prediction in combination with the reservoir, i.e., as part of the (linear) flow map, as we propose it here.
>
> [W2] In terms of the inverse operation, the complexity of the operation is indeed the number of neurons $M$ cubed. Which is the same complexity for computing the $C$ matrix. The difference between the two are $O(2M^3)$, instead of $O(M^3)$. This certainly adds some additional computational time, but in our experiments, we observe that these differences are negligible. In particular, they are not noticeable when compared to backpropagation.
>
> [W3] This is correct; it still does not provide the required number of neurons and is a hyperparameter that needs tuning. In the experiments, we observe that the required neurons to get good results are in the hundreds (which also means the addition of $K$ is not too computationally heavy, cf. previous weakness). In terms of theory, one has results with certain rates and probabilistic bounds for feedforward neural networks, but the issue is in adding this to the Koopman theory. We are still working on this, and we hope to understand better in the future.
>
> [W4] We thank the reviewer for pointing this out. There is indeed a mistake in the manuscript. When discussing one arbitrary time step, we consistently write $h_t$ to $F(h_t)=h\_{t+1}$. However, for the dataset, we use $h_n$ and $F(h_n) = h_n'$ to denote the nth data point and the resulting output after applying the evolution operator. This is because denoting the data as $\[h_1, ..., h_T\], h\_{t+1} = F(h_t)$, implies that the data is from one trajectory, which is not the case. The data consists of $N$ points $H$ and $N$ points $H'$ after applying the evolution operator once to $H$.
>
> The mistake was on line 143 and has been updated. Whenever we discuss the specific points in our dataset, we refer to $h_n$ and $h_n’$; for arbitrary points, we denote $F(h_t)=h\_{t+1}$.
>
> [Q1] These decisions were driven mainly by the modeling trends for certain tasks. For example, we consider ESNs to be prominent when it comes to modeling chaotic systems, while we consider LSTMs to be more typical for real-world multivariate time series forecasting with partial observation. In addition, LSTM is a known benchmark for this particular data.
>
> [Q2] Following equation 4, we construct the weight as the direction between the two points we sampled and then put the bias on the line between the original points. To set the bias between two points means that after the weight is applied, part of the points on the line between the two points are less than the bias, and the rest are greater than. This means we can capture the direction of the underlying manifold/input space and set the hyperplane (if using ReLU) or the zero line (tanh) in interesting areas of the manifold. And if the manifold/input space changes drastically, the weight space also adapts.
>
> For the distribution, we construct it by considering the input space and the output space. It emphasizes a pair of points close in the input space that is far away in the output space, such as where the gradient of the true function is high. This means the weights and biases are put in interesting areas of the manifold, where the function changes rapidly.
>
> If the data is not in a supervised setting, as with the controlled input $x$, we sample uniformly among the possible pair of points. This still considers the data through equation 4, but not to the same extent.

---

> > ### Comment · Reviewer_Bq2A · 2024-11-27
> >
> > Thanks for addressing my questions.  I stand by my rating.

---

### Official Review · Reviewer_vQoQ · 2024-11-04

**Soundness:** 3
**Presentation:** 3
**Contribution:** 2
**Rating:** 6
**Confidence:** 4

**Summary:**

The authors propose a novel way to construct recurrent neural networks. Their approach is two stepped, where they first generates hidden weights and biases according to a data-dependent distribution, and then construct read-out parameters by approximating the dimensional Koopman operator with dynamic mode decomposition.

**Strengths:**

Training recurrent neural networks is notoriously hard. I am partial to the authors approach of fitting an RNN to a a dynamical system by a smart sampling of hidden parameters and Koopman operator based modeling. Particularly, I am drawn to the use of Koopman operator in the context of RNNs.

The paper is very well written and is a nice read, and the results comparing their gradient-free approach to trained models are impressive.

**Weaknesses:**

It is unclear how much the Koopman operator and EDMD components contributed to model performance. Since the hidden weights initialisation schema is an application of previous work (Bolager et al. (2023)), I see the Koopman related work as the main conceptual innovation. However, based on the presented results it is hard to disentangle where improvements come from and I am not fully convinced of the added value of EDMD. I suggest the authors include two additional experiments: a. setting hidden weights randomly and learning read-out with EDMD and b. setting hidden weights based on Bolager et al. (2023) and learning read-out without EDMD.

I am also not keen on the name and think it over-promises. Being able to train general recurrent neural networks without gradient descent or BPTT is an extremely ambitious goal, which this paper does not fulfils. As the authors explain in the (very much appreciated) limitation section, their approach does not immediately extend to RNN tasks relating to computer vision or NLP, thus the paper results are mostly regard dimensional dynamical system. I still believe their results are impressive, but the language, and specifically the title, needs to be toned down.

**Questions:**

The impact of exploding and vanishing gradients is not made clear in this manuscript. I understand their model outperforms LSTMs and other, gradient based models based on numerical values. Could the authors make the impact of EVGP more apparent?

Can the authors include more real-world datasets? As of now, all but 1 result is simulated, and additional examples would significantly strengthen the manuscript.

---

> ### Author Response · Authors · 2024-11-23
>
> We would like to thank the reviewer for a great review, that has indeed improved the updated paper in our opinion. Below you will find a few remarks to the weaknesses you pointed out, as well as answer to your questions.
>
> [W1] We agree with the reviewer that such results should be added. We are still running experiments for this, and hope to return very soon with some updated results.
>
> [W2] This is a fair criticism, and we certainly are not intending on claiming more than we can show for; hence the limitation section as well. We have reviewed the title, and decided upon adding "for low dimensional data", which is what we demonstrate at the current moment. We still find the contribution very valuable, and look forward to tackle high dimensional tasks such as vision and NLP. We also added an additional nod to the limitation in the end of the introduction.
>
> [Q1] Training RNNs with gradient descent algorithms is notoriously hard, especially if the data are governed by long and complicated dependencies. In particular, classical RNNs trained to model chaotic systems, regardless of their architecture, will inevitably suffer from the exploding gradient problem \[1\]. Consequently, gradients grow exponentially during backpropagation, leading to extremely large weight updates. Thus, the model's parameters can diverge, causing training to become unstable, or the model may fail to converge, or it may be very difficult to find suitable hyperparameters, and so on.  Moreover, if gradients vanishes during backpropagation, the weight updates become negligible, making it difficult for the model to learn from the data, or the model struggles to learn long-term dependencies, as the influence of earlier time steps diminishes quickly, or the model may fail to capture complex patterns in the data, leading to underfitting. Therefore, EVGP results in RNNs becoming unable to learn the underlying temporal patterns.
>
> \[1\] Chaos paper: J. Mikhaeil, Z. Monfared, D. Durstewitz, On the difficulty of learning chaotic dynamics with RNNs, NeurIPS, (2022).
>
> [Q2] We have trained a sampled RNN with an additional real-world dataset, and added this to the appendix. Specifically we use a dataset containing household electric power consumption by UC Irvine (https://archive.ics.uci.edu/dataset/235/individual+household+electric+power+consumption), which was used in other publications working with real-world timeseries data (https://arxiv.org/pdf/1509.05142, https://proceedings.neurips.cc/paper_files/paper/2017/file/1b5230e3ea6d7123847ad55a1e06fffd-Paper.pdf).
> Further details and results are given in appendix I. We observe good performance for our method on this dataset as well.

---

> > ### Comment · Reviewer_vQoQ · 2024-11-23
> > **Response to authors**
> >
> > Dear authors,
> >
> > Thank you for response.
> >
> > Regarding W1, my main concern unfortunately still stands. I understand adding more experiments takes time and might be infeasible given the short rebuttal process of ML conference venues. However, since the RNN weights are direct implementation of previous work (Bolager et al. (2023)), the main novelty (in my eyes) of this manuscript is the Koopman operator for the read-outs. To this point, I am currently not convinced of its standalone merits.
> >
> > It is possible I have misunderstood parts of your work, and I am open to revising my score if the concerns are addressed.

---

> ### Comment · Reviewer_vQoQ · 2024-11-25
> **Response to authors**
>
> Dear authors,
>
> As I have stated in my original review and first response, in my view the main innovation of the work is using EDMD to train RNN output weights, as the initialization for the input and hidden weights is a **direct** application of [Bolager et al. (2023)](https://proceedings.neurips.cc/paper_files/paper/2023/file/c7201deff8d507a8fe2e86d34094e154-Paper-Conference.pdf). In my initial review, I requested evidence demonstrating EDMD's specific contribution to RNN training effectiveness compared to standard approaches. The authors' response did not provide experimental results addressing this core concern.
>
> Therefore, I recommend rejection and am lowering my score to a 3 from 5.

---

> > ### Author Response · Authors · 2024-11-25
> >
> > We understand your concerns, and we finally have the results you requested, as well as some remarks. We hope you are willing to have a look, and we apologize for the delayed response; it took some time to prepare all the results. Below you can see the results, where for the case "without SWIM", we sample from a Gaussian distribution for weights and uniform distribution for biases. We list the mean (min, max) error on the test set.
> >
> > | Experiment  | Without SWIM + With Koopman | With SWIM + Without Koopman | With SWIM + With Koopman   |
> > |-------------|-----------------------------|-----------------------------|----------------------------|
> > | Van der Pol | 2.90e-3 (6e-4,8.1e-3)       |  3.12e-2 (3.05e-2, 3.16e-2) | **9.55e-4 (7.08e-4, 1.28e-3)** |
> > | Lorenz      |  7.79e-3 (5.79e-3, 9.87e-3) |  8.41e-3 (6.45e-3, 1.02e-2) | **4.36e-3 (3.66e-3, 5.36e-3)** |
> > | Rössler     | 4.47e-1 (1.7e-4, 2.21e-0)   | 7.51e-3 (1.00e-4,3.63e-2)   | **1.57e-4 (5.86e-5, 3.82e-4)** |
> >
> > The new experiments show that adding SWIM and Koopman separately have a positive effect on performance in all cases. Also, interestingly, different systems benefit differently from adding SWIM and adding Koopman.
> >
> > In a separate experiment, we now also demonstrate that the performance on the approximation of Van der Pol dynamics when we increase the time step size consistently performs better than without EDMD and has a much better error bars (see plot in appendix J plus more details).
> >
> > We would like to emphasize that not only the performance is better in both new experiments, but also that with the addition of Koopman, we can apply linear control tools such as LQR on controlled examples (such as the forced Van der Pol in our paper), whereas we would need to resort to non-linear control when not using EDMD/Koopman. Linear control theory is much better understood and comes with a richer toolbox than nonlinear approaches. Hence, we think this is an additional, major benefit that should not go unnoticed.
> >
> > Separately, with the addition of Koopman theory, it gives a new interpretation of each component in the RNN, as well as allows for numerical analysis of the spectrum of the operator, something that we have performed in Appendix C.
> >
> > We want to take the opportunity to thank the reviewer for asking about the additional experiments, which turned out to be very interesting and helped strengthen our point. We also note that we have extended Appendix I with another real world data experiment.

---

> > > ### Comment · Reviewer_vQoQ · 2024-11-25
> > > **Response to authors**
> > >
> > > I commend the authors for adding these experiments in the relatively short time-frame of the ICLR rebuttal. I am now convinced of the merits of their approach. I will raise my score beyond the original 5 to a 6.

---

### Official Review · Reviewer_Jk3B · 2024-11-04

**Soundness:** 3
**Presentation:** 3
**Contribution:** 3
**Rating:** 5
**Confidence:** 3

**Summary:**

This paper presents an alternative strategy to backpropagation through time (BPTT) by using a combination of Koopman operator theory and random feature networks instead of gradient-based techniques. This novel method avoids vanishing and exploding gradient problems, and outperforms BPTT in terms of training time and accuracy in a series of empirical comparisons, including time series, forecasting, control, and weather problems.

**Strengths:**

1. Proof seems to be sound and theoretically correct given the assumptions stated by the authors, however i am not an very well versed with koopman operator theory and thus have provided a lower confidence score.
2. Their method (called sampled RNN) takes significantly less time to train than alternative state-of-the-art models such as ESNs, shPLRNNs, and LSTMs. Predictions from this model capture patterns in toy experiments as well as real-world data (like weather forecasting), and outperform current models (especially on problems with long horizons).

**Weaknesses:**

1. There is little motivation for the problem in the introduction, and the paper jumps right into formalisms.
2. The weights and biases need to be sampled from a data-dependent probability distribution, however it's unclear how feasible this is?
3. This method does not converge for controlled systems.

**Questions:**

1. “For completeness we have added sigma_hx as an arbitrary activation function. We choose to set sigma_hx as the identity function to let us solve for the last linear layer… other activation functions such as the logit are possible as well”. However this is not shown in the paper. Can the authors provide clarification on the effect of non-linear activations here?
2. In the weather task, the sampled RNN has a similar MSE to shPLRNNs and LSTMs for 1-day forecasting, but higher than both alternative models for week-long forecasting. However, MSE is higher than ESNs for the Rosseler system task. Can the authors provide some clarification on this?
4. Can the authors provide clarifications/results on how performance for chaotic systems might change based on how many samples are drawn for the sampled RNN (and the influence on dimensionality of the hidden layer)?
5. The authors state: “The complexity of solving this system depends cubically on the minimum number of neurons and the number of data points (respectively, time steps). This means if both the network and the number of data points grow together, the computational time and memory demands for training grow too quickly. For BPTT, the memory requirements are mostly because many gradients must be stored for one update pass”. — can the authors provide a comparison of # of neurons vs # of time steps?
6. Can the authors provide details on how they infer the “data-dependent probability distribution” (especially for the weather data)?
5. Additionally, can the authors expand on how weights and biases are sampled from the specific, data-dependent probability distribution (again for weather data) and how other parameters are computed using a sequence of linear equations?

---

> ### Author Response · Authors · 2024-11-23
>
> Thank you for the insightful review, which we believe have strengthened the updated paper. Below we make some thoughts and remarks on the weaknesses you point out, as well as answers to your questions.
>
> [W1] Our thinking behind the introduction is to quickly point out some known problems with training RNNs before giving a quick intuition of our final model. We then enter a more extensive related work section, contextualizing and motivating the problems we quickly mention in the introduction. However, we agree with the reviewer that the change of pace is perhaps a bit too rapid, and we have rewritten part of the introduction to give some more motivation and a less formal introduction to the method we will describe in the paper.
>
> [W2] We use two different data-dependent probability distributions, briefly described around line 182. They are chosen based on whether we work in a supervised setting. As the latter distribution is simply a uniform distribution over the input data, this is always possible to derive. This is still a data-dependent probability, as it still captures the underlying properties of the input space. That is, two different input spaces create two different weight/bias spaces, meaning the distribution we sample from differs.
>
> [W3] It is true it is harder to show convergence for controlled systems, which we briefly discuss in Appendix B.2. This is also known as a problem in the Koopman community. However, this is when we show convergence given the specific way we estimate K, which means we are also saying something about the learning algorithm. This is even stronger as a result, as it holds for arbitrary finite horizon prediction, without saying we need arbitrary long trajectories as data. If one merely talks about existence and does not consider the algorithm nor the data, then known results of universal approximation of RNNs can be applied. That being said, we want to understand better convergence for controlled systems, also theoretically, and not just demonstrate it experimentally as we do in the paper.
>
> [Q1] If the state space we map to differs from the Euclidean space, e.g., $[0,1]^d$, then we may choose $\sigma_hx$ to map from the Euclidean space and down to $[0,1]^d$. For the specific example, the final approximation is changed from linear to logistic regression. This means we need an iterative method such as gradient descent, but we only need to compute gradients for the final layer; hence, there is no backpropagation, which is still faster than BPTT. Of course, if one can formulate the approximation of the outer layer as a linear regression problem, as we do in the rest of the paper, it is preferred because it is much faster and gradient-free.
>
> [Q2] There is often a trade-off between accuracy and training time, and our goal is to minimize this and even be better when possible.
>
>  More concretely, for chaotic systems, we would like to clarify that we did not use MSE as a measure of error for chaotic systems like the Rössler system; instead, we utilized EKL (empirical Kullback-Leibler divergence). Note that no single evaluation metric may offer a complete picture for assessing model performance for complex or chaotic data. As noted in \[1\], one may consider multiple evaluation metrics to evaluate model performance, particularly in chaotic or complex data. This paper provides several metrics, such as D_stsp, D_H, and PE, to evaluate model performance. In some cases, it is probable that while one or two metrics favor a particular method, others may indicate higher errors (for instance, see Table 1 in \[1\]). With this in mind, in the case of the Rössler example, we acknowledge that the EKL for our method is slightly higher than that of ESNs. However, we would argue that the EKL between the models is fairly similar, and we can conclude that the performance is comparable, as well as our method being faster than the other methods.
>
> One reason the LSTM and shPLRNN produce better MSE for the weather data could lie in their predictions’ (in)stability. As shown in Figure 6 of the paper, the LSTM model doesn't capture fluctuations in the data, while shPLRNN and sampled models do. We can also see that fluctuations of the sampled model have a higher frequency than those of shPLRNN. Predicting higher frequencies is a good strategy when forecasting a short horizon, but it can lead to overfitting on a longer horizon. Thus, we can possibly see these results as underfitting from the LSTM and overfitting from the sampled model in the case of week-long forecasting.
>
> \[1\] GTF paper: F. Hess, Z. Monfared, M. Brenner, D. Durstewitz, Generalized Teacher Forcing for Learning Chaotic Dynamics, ICML, (2023).
>
> To answer all your questions adequately, we continue below and hope the reviewer finds this acceptable. Thanks!

---

> > ### Author Response · Authors · 2024-11-23
> >
> > [Q3] We explored various hidden layer dimensionalities as an ablation study for the chaotic datasets. First we want to note that since weights and biases are constructed using sampled pairs from the dataset, this means that the number of model parameters is proportional to the number of considered datapoints (i.e. the model possibly does not see the full dataset).
> > We observed that the performance improves with an increased number of parameters, possibly saturating or overfitting after a certain point. This was also added in the appendix, and we refer to lines 1437 - 1457 for this.
> >
> > [Q4] The computationally heavy part is the inverse operation, the complexity of the operation is indeed the number of neurons $M$ cubed (or number of data points if we have more neurons than data). In our experience, the number of neurons required are in the hundreds, and is not a computational burden. If we require much, much more, we luckily have a lot of computationally efficient solvers, as linear regression and least square solves are very well studied (although the technical details of such solvers is beyond the scope of the paper). In addition, the complexity of our method is not directly affected by number of time steps, such as backpropagation-through-time, only in so far as it increases the number of data points when we apply the least square solver.
> >
> > [Q5] We do the following for the weather data and every other example we provide. We first perform subsampling by uniformly sampling points from the training set, which includes both the input h and the true output F(h). Then, for all the pairs in this smaller dataset, we evaluate the difference between the output divided by the difference of the input (briefly described in the paper around line 182). These values are then used as the probability mass function in a discrete distribution to sample the pair of points.
> >
> > Informally, this means we want to sample a pair of points close in the input space but far away in the output space, i.e., where the gradient of the true function is high, as this is most likely a good place to put weights and biases around. This implies that the data "decides" how the probability distribution looks, hence data-dependent probability distribution.
> >
> > [Q6] For the weather data and all the other examples, we sample $M$ pair of points where $M$ is the number of neurons we have. The sampling is done according to the previous answer. Then, we construct each weight and bias according to equation 4 (line 169 in the paper). As the weights/biases are constructed only by the data, the weights and biases are data-dependent parameters.
> >
> > Informally, we construct the weight as the direction between the two points we sampled and then put the bias between the two points. To set the bias between two points means after applying the weights one half of the points on the line between the two points are less than the bias, and half is greater than. This ensures that the weight direction and the bias is following the underlying manifold/input space, and changes if the manifold/input space changes.

---

### Official Review · Reviewer_hnnr · 2024-11-13

**Soundness:** 4
**Presentation:** 2
**Contribution:** 2
**Rating:** 5
**Confidence:** 5

**Summary:**

-  The paper proposes a training method for modeling recurrent neural networks without the use of gradient-based methods, such as backpropagation through time (BPTT), which suffers from exploding and vanishing gradients, mostly occurring in a system with chaotic dynamics. Building on concepts from random feature models, such as reservoir computing (echo-state networks), the paper proposes the random sampling of weights (W­­­ and b) in the RNN from a data-driven distribution. In addition the paper employs Koopman operator theory to find the outer weights of the RNN model, which map the current state to the next state. The Koopman operator theory maps the finite nonlinear transformation matrices (outer weights) to a linear infinite-dimensional space, where the extended dynamic mode decomposition (EDMD) method is used to find a finite-dimensional approximation of the Koopman operator.
-  For model validation, they show some computational experiments comprising simple ODEs, such as the Van Der Pol Oscillator, chaotic dynamics (Lorenz and Rossler systems), and real-world examples involving weather data.
- Paper reports results from these computational experiments in the form of training time and error (MSE/KL Divergence). When compared to other models, such as an LSTM, ESN (echo-state network), and shPLRNN (state of the art backpropagation-based RNN), the proposed model (Sampled-RNN) achieves comparable performance, in terms of MSE and KL Divergence, and a faster training time.

**Strengths:**

- Interesting connection to Koopman operator.
- Interesting topic of trying to circumvent gradient based training.

**Weaknesses:**

- Sampling procedure not clearly explained. (E.g. 'As we stick to networks with one hidden layer in this paper, we ignore the multilayer sampling here and direct the reader to Bolager et al. (2023) for the full sample and construction procedure for an arbitrary number of hidden layers.')
- Paper presentation generally not clear. Hard to follow. Illustrative example: equation 1 is referred to before presented. Paper requires excessive ‘detective work’ to understand what they did and/or are talking about.
- How sampling approach differs from ESNs seems unclear, and a minor innovation at best.
- Although interesting, connection to Koopman operator theory does not seem novel.

**Questions:**

N/A

---

> ### Author Response · Authors · 2024-11-23
>
> We thank the reviewer for the review which we believe have improved our new version of the manuscript. Below we add a few comments on their weakness points:
>
> [W1] We chose to refer to the original paper for multilayer as we are not using more than one hidden layer for each network we sample. We still provide the details for how a pair of points form the weights and bias and how we sample them by describing two probability distributions on line 183. We still recognize that it certainly is not easy to briefly understand the complete algorithm.
>
> [W2] Thank you for pointing this out. We agree that, in particular, the introduction can be hard to follow. To make it clearer and avoid jumping straight into formalism, we have rewritten part of the introduction to alleviate the issue.
>
> [W3] The sampling approach approximates the Koopman map, which would be a random matrix for an ESN. This comes with three benefits: 1) increased interpretability and better tools to analyze the network --- both theoretically but also through spectral properties of the approximated Koopman matrix, and 2) significantly decrease in the hyperparameter search space, 3) the possibility of including linear control tools such as LQR (see forced Van der Pol in paper).
>
> Furthermore, we consider the data-dependent sampling distribution to be a key element in our method, which can reduce the number of necessary hidden units, as well as less hyperparameter tuning, and thus a shorter training time compared to ESNs. We confirmed this in numerical experiments.
>
> [W4] There is a lot of research on numerical approximation of the Koopman operator, including papers on random Fourier features used as dictionary in EDMD (as cited in the related work).
>
> However, there is very little research that connects recurrent neural networks with Koopman operator theory. Our paper provides one of the first theoretical results in this direction - and we are not aware of any paper that directly ties recurrent network weights to Koopman operator approximations.
>
> We also refer to our response to reviewer 4 here, because it provides related information:
> The connection between Koopman operator approximation methods and reservoir computing (RC) / echo-state networks (ESNs) has not been explored extensively. We are only aware of the two papers we cite in the section on related work (Bollt (2021); Gulina & Mauroy (2020)). Bollt (2021) only briefly mentions the connection of RC to DMD (i.e. without nonlinear observables), while Gulina & Mauroy (2020) use the hidden state of the reservoir as a dictionary to approximate the Koopman operator - but do not use the Koopman operator for prediction in combination with the reservoir, i.e., as part of the (linear) flow map, as we propose it here.

---

> > ### Comment · Reviewer_hnnr · 2024-11-30
> >
> > Thank you to the authors for their response. However, after careful consideration, I must stand by my original review and maintain that this work is not yet ready for publication. Upon rereading the submission, I now see the connection to the Koopman operator as the primary innovation of this work. Even with the additional experiments addressing reviewer vQoQ's concerns, I do not find these innovations sufficiently significant.
> >
> > Furthermore, the paper presents these ideas in a “bundled” manner, combining them with concepts that feel tangential, such as the reuse of ideas from Bolager (2023). This framing—particularly the portrayal of this work as a solution to the EVGP—creates unnecessary confusion and obfuscates the contribution. A substantial portion of the work appears to derive from Bolager, which further complicates the narrative.
> >
> > If the key contribution is the application of Koopman operator theory to RNNs, I recommend that the authors focus exclusively on that innovation, independent of their trajectory (e.g., their investigations into sampling RNN-based methods for network training). For instance, as I understand it, nothing prevents the authors from applying Koopman operator theory to RNNs trained via backpropagation. While this does not invalidate the current results (combining Koopman theory with sampling), it suggests that the paper only tells part of the story, in other words, this work is not complete. Moreover, it presents the relationship between sampling-based RNN training and Koopman operator theory as deeper than it appears to be in a way that may simply lead to unnecessary confusion within the community.
> >
> > I commend the authors for their novel, but not yet matured, work but stand by my view that this interesting approach is not yet developed enough and not presently presented in a way that is likely to be a strong contribution to ICLR.

---

> > > ### Author Response · Authors · 2024-12-04
> > >
> > > We thank the reviewer for their time and this rebuttal session, and would like to add a few comments on their conclusion. As the reviewer points out, this paper presents several ideas combined. Importantly, the parts are necessary and complement each other. We highlight the component’s reliance on each other below.
> > >
> > > - Using gradient methods to optimize and approximate the Koopman operator results in a much poorer approximation than the EDMD algorithm. The EDMD algorithm requires a fixed basis, which can be achieved by sampling the hidden layer first and keeping it fixed. This cannot be achieved as accurately and as fast through backpropagation-through-time (or other gradient methods) compared to a linear solver.
> > > - Sampling without the addition of Koopman also leads to worse results, which can be seen in Appendix J. Hence, the method really shines in the combination of sampling and Koopman.
> > > - As a result, we do indeed relieve all issues with EVGP in the setting we consider, and we think this is part of the main motivation of the paper and the one we set out to solve. This also leads to a much faster optimization time and more tools to analyze and interpret the network. These are all things we think are worth considering and answering.
> > >
> > > So overall, even though the Koopman connection is indeed an important one, and we acknowledge that theoretically, one can work without the sampling addition, computationally, it is much more advantageous to combine the two.
> > >
> > > Finally, the paper does indeed deliver several ideas and new tools - as the reviewer also points out - which, in our view, is a necessary combination and makes the paper interesting for a larger audience, whether they come from a computational, explainable AI, or dynamical systems background.

---

### Meta-Review · Area_Chair_YtA5 · 2024-12-21

**Metareview:**

In this work, the authors propose a gradient-free approach to training recurrent neural networks in low-dimensions. The proposed approach applies EDMD for output weight training to the sampling RNN framework of Bolager et al. (2023), bypassing backpropagation, and negating the exploding-vanishing gradient problem. Reviewers agreed that the manuscript is well-written, and that the approach is sound and yields good results.

The more negative responses pointed out that the primary contribution was the incorporation of EDMD for the training procedure. The use of sampling strongly resembles ESNs, which can also be considered "gradient-free" in the same way as the proposed method. From this viewpoint, the new contribution pertains to a difference in architecture in the final layer, which requires the EDMD approach. This is not how the method is portrayed, however, as has been pointed out by reviewers. I disagree with this as a minor issue, or easily addressed by further experiments, as the exposition should be upfront about this, and the title not so broad to compensate. It is also strange that ESNs were not compared to in all settings. The authors commented on this in the discussion period, claiming that ESNs are best for chaotic systems, but if so, why not include them? Finally, I am surprised that reviewers did not critique the baselines, as I would not consider shPLRNNs, LSTMs or ESNs to be state of the art.

I am uncomfortable with recommending the paper for acceptance as-is, and would instead suggest that the authors think more deeply about the relationships between their proposed method and other existing approaches, including ESNs and the actual state-of-the-art, perhaps including other RNN approaches inspired by Koopman operators and DMD.

**Additional Comments On Reviewer Discussion:**

Reviewer hnnr raised concerns about how the application of EDMD and sampling neural networks are sold as a "bundle", when the former aspect is really the primary contribution. This concern was also echoed by Reviewer vQoQ who recommended two additional experiments, where read-out is learned without EDMD, and the hidden weights are assigned using a more standard distribution. Impressed by these new results, Reviewer vQoQ increased their score. Reviewer Bq2A gave a highly positive review, but did comment on the unusual absent baselines. Reviewer Jk3B raised a few assorted questions, but did not engage in the discussion period.

---

### Decision · Program_Chairs · 2025-01-22

Reject